# CROSS-EMBODIMENT OFFLINE REINFORCEMENT LEARNING FOR HETEROGENEOUS ROBOT DATASETS

**Haruki Abe**[1,2]**, Takayuki Osa**[2]**, Yusuke Mukuta**[1,2]**, Tatsuya Harada**[1,2]
[1]The University of Tokyo, Tokyo, Japan
[2]RIKEN Center for Advanced Intelligence Project, Tokyo, Japan
`{abe,mukuta,harada}@mi.t.u-tokyo.ac.jp, {takayuki.osa}@riken.jp`

## ABSTRACT

Scalable robot policy pre-training has been hindered by the high cost of collecting high-quality demonstrations for each platform. In this study, we address this issue by uniting offline reinforcement learning (offline RL) with cross-embodiment learning. Offline RL leverages both expert and abundant suboptimal data, and cross-embodiment learning aggregates heterogeneous robot trajectories across diverse morphologies to acquire universal control priors. We perform a systematic analysis of this offline RL and cross-embodiment paradigm, providing a principled understanding of its strengths and limitations. To evaluate this offline RL and cross-embodiment paradigm, we construct a suite of locomotion datasets spanning 16 distinct robot platforms. Our experiments confirm that this combined approach excels at pre-training with datasets rich in suboptimal trajectories, outperforming pure behavior cloning. However, as the proportion of suboptimal data and the number of robot types increase, we observe that conflicting gradients across morphologies begin to impede learning. To mitigate this, we introduce an embodiment-based grouping strategy in which robots are clustered by morphological similarity and the model is updated with a group gradient. This simple, static grouping substantially reduces inter-robot conflicts and outperforms existing conflict-resolution methods.

## 1 INTRODUCTION

In recent years, breakthroughs across many areas of machine learning have been driven by scaling up both the size of the model and the dataset. In particular, large language models (LLM) and vision-language models (VLM) pre-trained on the web-scale, diverse and massive data have become capable of solving a wide variety of linguistic and visual tasks (OpenAI, 2023; Gemini Team, 2023). Likewise, generative models for image and video generation, as well as music generation, can now produce outputs of unprecedented quality (Rombach et al., 2022; Singer et al., 2022).

This trend has also begun to influence robotics, where scaling transformer-based architectures and training them on large, heterogeneous robot datasets have produced "robot foundation models" capable of multiple tasks within a single model (Black et al., 2024; Open X-Embodiment Collaboration, 2024; Octo Model Team et al., 2024; Kim et al., 2024). However, despite the promise of foundation models for robotics, they face a critical limitation. Available demonstration data remain extremely limited compared with the vast text and image corpora that fuel today's foundation models. Collecting manipulation data is time-consuming and expensive, and each new task requires careful teleoperation, specialized hardware, and often manual labeling, making data scaling difficult.

To overcome this data bottleneck, researchers have turned to cross-embodiment learning, where a single model is pre-trained on demonstrations from many different robot platforms rather than just one. By pooling heterogeneous data, the model can learn more generalizable control primitives. Building on this, the key practical benefit is adaptability, because a cross-embodiment pre-trained policy can be adapted to new robots with only modest additional data, avoiding per-robot data collection and from-scratch training and thereby lowering costs and speeding deployment. Recent VLA work supports this direction, showing that parameter-efficient and optimized fine-tuning can achieve strong performance and substantial speedups with limited data (Kim et al., 2024; 2025).

However, these foundation models for robotics have so far been confined to imitation learning, which requires high-quality demonstration data, typically gathered via expert teleoperation, and therefore remains costly for each individual robot. To overcome this limitation, there is growing attention on offline reinforcement learning (offline RL) as a pre-training method. Offline RL can leverage not only expert demonstrations but also suboptimal data, enabling policy learning that can surpass pure imitation learning in settings where high-quality data are difficult to acquire, such as robotics (Levine et al., 2020). For example, Q-Transformer (Chebotar et al., 2023) demonstrates that, when offline RL is applied to large-scale robotic datasets containing suboptimal trajectories, the model can outperform behavior cloning (BC), one of the most widely used approaches to imitation learning. This suggests a natural complementarity. By combining cross-embodiment learning with offline RL, one could exploit heterogeneous robot data and both expert and suboptimal data to dramatically expand the pool of data usable for pre-training. However, this promising combination remains largely unexplored, and there is no established methodology or systematic analysis.

The contribution of this paper is an empirical study that systematically investigates the challenges and potential of cross-embodiment offline RL. We present a new benchmark and analysis of pre-training combining offline RL with cross-embodiment learning on data collected from up to 16 different robot platforms. Our experiments reveal two key findings: (i) under conditions with substantial suboptimal data, the combined cross-embodiment and offline RL approach outperforms imitation learning, and (ii) as the proportion of suboptimal data and the number of robot types increase, improving performance becomes difficult for particular embodiments under naive offline RL. We analyze this phenomenon and find that, in cross-embodiment learning, each robot's gradients conflict and cause some robots' gradients to vanish. This phenomenon leads to a degraded performance. To address this, we propose a novel group-task update strategy based on robot embodiment information. We encode each robot as a morphology graph, compute pairwise distances, and cluster robots accordingly. We then update the policy group by group, sequentially applying an actor update using only the samples of the group. This grouping mitigates gradient conflicts and yields substantial performance gains in settings with high suboptimal data ratios where standard offline RL stalls.

> This paper makes the following contributions:
>
> 1. We introduce and analyze the new benchmark that combines offline RL with cross-embodiment learning across up to 16 distinct robot platforms.
>
> 2. We show that offline RL with cross-embodiment outperforms BC when suboptimal trajectories dominate, and that the pre-trained model can learn representations that markedly accelerate fine-tuning on previously unseen robots.
>
> 3. We find positive transfer among morphologically similar robots, while demonstrating that increasing the suboptimal-data ratio and the number/diversity of robots amplifies inter-robot gradient conflicts, which in turn induces negative transfer.
>
> 4. To mitigate these conflicts, we propose a simple, static grouping strategy that represents each robot as a morphology graph and clusters robots by graph-based distances; performing group-wise policy updates reduces gradient interference and yields large gains, up to 39.8% improvement, especially under high suboptimal-data conditions.

## 2 RELATED WORKS

### 2.1 OFFLINE RL

Offline RL aims to learn a policy that maximizes cumulative reward using only a static dataset of environment interactions without further online interaction (Levine et al., 2020). Compared to BC, which simply mimics the trajectories of the dataset, offline RL can stitch together suboptimal segments to produce better performance when the dataset contains a mix of expert and suboptimal data (Kostrikov et al., 2021; Kumar et al., 2020). For example, implicit Q-learning (IQL) (Kostrikov et al., 2021) first fits a state value function $V_\psi(s)$ via expectile regression to capture an upper expectile of the return distribution; it then uses the learned $V_\psi(s')$ as the TD target for updating a state–action value function $Q_\theta(s, a)$. Finally, the policy $\pi_\phi(a \mid s)$ is extracted via advantage-weighted BC, avoiding any need to evaluate out-of-distribution actions. This pipeline enables IQL to leverage sub-

optimal trajectories effectively and achieve strong performance without explicit policy constraints. In this work, we adopt offline RL as our pre-training paradigm to broaden robotic data, especially suboptimal data that can be exploited for foundation model learning in robotics.

## 2.2 CROSS-EMBODIMENT LEARNING

Cross-embodiment learning trains a single network on data collected from multiple robot morphologies, enabling transfer of control priors across platforms. Since collecting large datasets for any single robot is costly, pre-training on heterogeneous robot data has become a popular strategy to improve generalization capability (Open X-Embodiment Collaboration, 2024; Octo Model Team et al., 2024; Black et al., 2024; Kim et al., 2024). However, existing cross-embodiment foundation models rely almost exclusively on imitation learning (Open X-Embodiment Collaboration, 2024; Octo Model Team et al., 2024), and there has been little work combining cross-embodiment pre-training with offline RL. While Nakamoto et al. (2024) applied offline RL to data from two robot platforms, they did not analyze any cross-embodiment effects. Similarly, Springenberg et al. (2024) conducted offline RL on a dataset comprising two manipulators and several toy tasks but did not investigate the benefits or challenges of learning from many distinct embodiments simultaneously. To fill this gap, we introduce the new benchmark that systematically combines offline RL with cross-embodiment learning, analyze the interactions between these paradigms, and propose methods to mitigate the challenges that arise when pooling heterogeneous and often suboptimal robot data.

## 3 EXPERIMENTAL SETUP

### 3.1 PROBLEM SETTING

We study multi-embodiment offline RL, where a single policy must control multiple robot morphologies under a common state–action interface. Unlike the standard multi-task RL setting, where a single robot embodiment solves multiple tasks with different rewards or goals, here multiple robot embodiments solve a common locomotion objective and reward functions that share the same components but may use embodiment-specific weights. Such settings have been referred to as cross-embodiment or multi-embodiment learning (Open X-Embodiment Collaboration, 2024; Bohlinger et al., 2024). In this work, we follow this terminology and refer to our setting as cross-embodiment offline RL. Concretely, let $\mathcal{T}$ denote a finite set of robot embodiments (e.g., different quadrupeds, bipeds, hexapods) and let $f^{\mathrm{morph}} : \mathcal{T} \to \mathbb{R}^{d_m}$ map each embodiment index $\tau$ to a morphology descriptor $f^{\mathrm{morph}}(\tau)$. Each embodiment $\tau \in \mathcal{T}$ induces an MDP $\mathcal{M}_\tau = \left(\mathcal{S}, \mathcal{A}, p_{0,\tau}(s), P_\tau(\cdot \mid s, a), r_\tau(s, a), \gamma\right)$ where $\mathcal{S} \subset \mathbb{R}^{d_s}$ is the shared observation space (joint angles, velocities, etc.), $\mathcal{A} \subset \mathbb{R}^{d_a}$ is the continuous control space, $p_{0,\tau}(s)$ is the embodiment-specific distribution over initial states, $P_\tau(s' \mid s, a)$ is the embodiment-specific environment dynamics, $r_\tau(s, a)$ is a dense embodiment-specific reward and $\gamma \in (0, 1)$ is the discount factor. Episodes may end according to a terminal condition encoded in the transition tuples through a termination indicator $d_t \in \{0, 1\}$.

We assume access to a pooled offline dataset $\mathcal{D} = \bigcup_{\tau \in \mathcal{T}} \{(s_t, a_t, s_{t+1}, r_t, d_t)\}_{t=1}^{N_\tau}$ generated by an unknown behavior policy $\pi_\beta(a \mid s, f^{\mathrm{morph}}(\tau))$. Our goal is to learn a single parameterized policy $\pi_\theta(a \mid s, f^{\mathrm{morph}}(\tau))$ that maximizes the expected cumulative discounted return over all embodiments $\mathcal{R} = \sum_\tau \sum_{t=0}^{T} \gamma^t r_\tau(s_t, a_t)$, using only $\mathcal{D}$. Rather than providing the policy with a one-hot embodiment index, we rely on morphology features (size, mass, link lengths, etc.), which we formalize as the descriptor $f^{\mathrm{morph}}(\tau)$. In this way, the same policy $\pi_\theta(a \mid s, f^{\mathrm{morph}}(\tau))$ can generalize across embodiments by conditioning on these universal state features.

### 3.2 ENVIRONMENTS AND DATASET

To facilitate cross-embodiment pre-training under an offline RL paradigm, we constructed a new locomotion dataset within the MuJoCo (Todorov et al., 2012) simulation environment, following the walking tasks of Bohlinger et al. (2024). Our dataset includes 16 distinct robot platforms, including nine quadrupeds, six bipeds, and one hexapod, each trained by Proximal Policy Optimization (PPO) (Schulman et al., 2017). During training, we record the tuple $(s_t, a_t, s_{t+1}, r_t, d_t)$ at each time step to capture the state, action, next state, reward, and termination signals.

For each robot, we curate six variants of 1 M–step datasets, divided by data quality:

- **Expert data**: 1 M steps collected by rolling out the fully converged PPO policy.
- **Expert Replay data**: all interaction steps from training start until expert-level performance ($\sim 500\,\mathrm{M}$ steps in total), uniformly subsampled to $1\,\mathrm{M}$ steps to bound dataset size.
- **70% Suboptimal Replay data**: $700\,\mathrm{k}$ steps drawn from the early (suboptimal) phase of PPO training, mixed with $300\,\mathrm{k}$ steps from the late (expert-like) phase, totaling 1 M steps.

Each of these Expert, Expert Replay, and 70% Suboptimal Replay datasets is provided in two walking-direction variants, Forward and Backward. In the Forward variant, the robot is commanded to walk forward at $1\,\mathrm{m/s}$, whereas in the Backward variant the commanded base velocity is $-1\,\mathrm{m/s}$. The reward in both cases is the same dense locomotion reward, and the tasks differ only in the commanded walking direction. See Appendix B for dataset construction details and Appendix C for reward distributions.

### 3.3 NETWORK ARCHITECTURE

In this section, we present our approach to cross-embodiment learning in an offline RL setting. The central challenge is to train a single network across robots whose state and action dimensions differ. To address this, we adopt the URMA architecture (Bohlinger et al., 2024), which enables multiple robots to share a single policy and state value function. URMA factorizes each observation into an embodiment-agnostic general part $o_g$ and a robot-specific part. For locomotion, the robot-specific stream is further split into variable-length sets of joint and foot observations, $\{o_j\}_{j \in J(\tau)}$ and $\{o_f\}_{f \in F(\tau)}$. Descriptor-conditioned attention aggregates each set into fixed-size latents, which are concatenated with $o_g$ to form a morphology-agnostic core representation. To facilitate offline RL, we further extend URMA by introducing a state–action value function ($Q$-function). Specifically, we encode each action with an action encoder to obtain a latent action vector, which we then concatenate with the latent representation of the URMA encoder. See Appendix D for architectural details.

## 4 ANALYSIS OF STANDARD OFFLINE RL ALGORITHM IN CROSS-EMBODIMENT OFFLINE RL

### 4.1 COMPARISON OF BEHAVIOR CLONING AND OFFLINE RL

Here, we compare BC, widely used in cross-embodiment training of foundational robot models, with offline RL. To date, applications of offline RL to robot foundation models have been rare, owing to the difficulty of learning from unlabeled interaction data; thus, a rigorous evaluation is necessary. Table 1 reports the performance of BC and implicit Q-learning (IQL) (Kostrikov et al., 2021), the commonly used offline RL method. On datasets with relatively uniform behavioral quality (for example, Forward Expert and Backward Expert), BC and offline RL achieve comparable results. In contrast, on datasets containing predominantly suboptimal trajectories, specifically 70% Suboptimal Replay Forward and Expert Replay data, offline RL methods surpass BC. This finding mirrors the results reported on benchmarks like D4RL (Fu et al., 2020) and confirms that offline RL remains robust even when datasets include significant suboptimal data in a cross-embodiment context.

Table 1: BC vs. IQL performance across datasets (mean $\pm$ standard error over 5 seeds).

| Dataset | BC | IQL |
|---|---|---|
| Expert Forward | $63.31 \pm 0.10$ | $63.39 \pm 0.05$ |
| Expert Backward | $67.17 \pm 0.01$ | $67.10 \pm 0.01$ |
| Expert Replay Forward | $49.71 \pm 1.06$ | $54.61 \pm 0.12$ |
| Expert Replay Backward | $42.87 \pm 1.32$ | $51.86 \pm 1.56$ |
| 70% Suboptimal Forward | $30.52 \pm 3.10$ | $36.62 \pm 1.02$ |
| 70% Suboptimal Backward | $41.42 \pm 0.71$ | $38.69 \pm 0.89$ |
| Mean | 49.17 | 52.05 |

### 4.2 EFFECTS OF CROSS-EMBODIMENT PRE-TRAINING

In the next experiment, we now evaluate how cross-embodiment pre-training impacts the performance of single-embodiment fine-tuning with offline RL. Figure 1 shows the learning curves for a

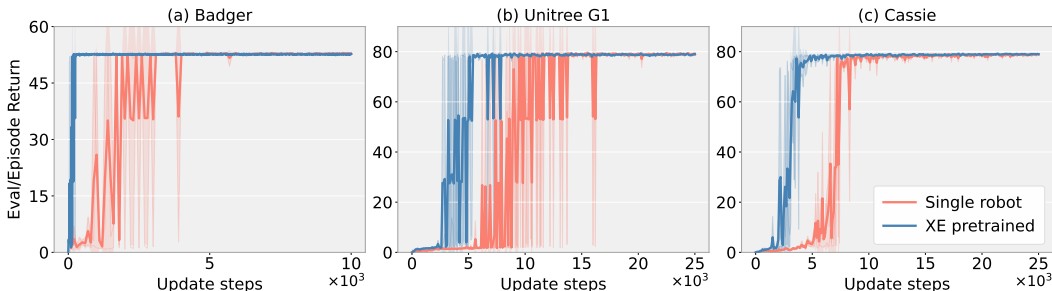

Figure 1: Comparison of learning curves between cross-embodiment pre-trained networks and networks trained without cross-embodiment pre-training for Badger, Unitree G1, and Cassie.

"leave-one-out" experiment. We pre-train via offline RL on a dataset excluding one robot, then fine-tune that robot with pre-trained networks, comparing it to a model trained without cross-embodiment pre-training. The model pre-trained with cross-embodiment data converges markedly faster. The results show that, for the quadrupedal robot Badger as well as for the bipedal robots Unitree G1 and Cassie, the network pre-trained via a cross-embodiment dataset is able to learn an effective policy more rapidly. These results demonstrate that cross-embodiment learning serves as a highly effective pre-training strategy in offline RL.

## 4.3 POSITIVE AND NEGATIVE TRANSFER IN CROSS-EMBODIMENT LEARNING WITH SUBOPTIMAL DATA

In this section, we first compare the final performance of models trained on each robot in isolation with a model trained via cross-embodiment learning. Table 2 shows the final rewards achieved by IQL when trained via cross-embodiment on the Expert Forward and 70% Suboptimal Replay Forward datasets, along with the rewards obtained by models trained separately on each robot.

In the Expert Forward dataset, the cross-embodiment models learn just as effectively as the single-robot models. However, an examination of the 70% Suboptimal Replay Forward dataset, which contains a large proportion of suboptimal data, reveals a different result. Although the average performance of cross-embodiment models falls below that of isolated models, certain quadrupedal robots, Unitree A1, Unitree Go1, Unitree Go2, and Badger, show substantial performance gains. This suggests that positive transfer occurs among the quadrupeds, likely because they contribute the largest share of data to the dataset.

By contrast, bipedal robots such as Unitree H1 and Unitree G1, which have relatively little similar-embodiment data in the dataset, suffer pronounced performance degradation under cross-embodiment learning compared to their isolated training. Because this negative transfer does not appear when using suboptimal data without cross-embodiment or when training on the Expert dataset which contains fewer suboptimal trajectories, we conclude that negative transfer emerges only when a dataset combines large amounts of suboptimal data with the cross-embodiment training regime.

Together these results show that the combination of cross-embodiment learning and offline RL can yield both positive and negative transfer. Negative transfer is most likely when suboptimal trajectories dominate and embodiment diversity is high. These findings indicate that effective cross-embodiment learning requires methods that minimize negative transfer while maximizing positive transfer.

## 5 GRADIENT CONFLICTS AND RESOLUTION VIA EMBODIMENT-BASED GROUPING

In this section, we analyze why combining large amounts of suboptimal data with cross-embodiment learning fails to yield performance improvements. Our empirical results indicate that gradient conflict is a primary cause of negative transfer. To address this issue, we propose a novel mitigation strategy that groups robots according to their embodiment, thus reducing gradient conflicts.

Table 2: Expert vs. 70% Suboptimal IQL performance across robots and avg. gradient cosine similarity $C$ on the 70% suboptimal dataset. Cells shaded blue ( ) indicate *large positive transfer* (CE exceeds Single by $> 10$), while cells shaded light red ( ) indicate *large negative transfer* (CE falls below Single by $> 10$).

| Robot | Exp Single | Exp CE | 70% Single | 70% CE | Avg. $C$ |
|---|---|---|---|---|---|
| unitree_a1 | 53.78 | 53.69 | 14.55 | 27.38 | 0.114 |
| unitree_go1 | 54.03 | 54.05 | 14.46 | 40.05 | 0.118 |
| unitree_go2 | 53.50 | 53.54 | 13.76 | 52.39 | 0.102 |
| anymal_b | 49.84 | 49.73 | 48.00 | 47.81 | 0.070 |
| anymal_c | 44.46 | 44.46 | 43.63 | 42.73 | 0.082 |
| barkour_v0 | 46.28 | 46.68 | 46.25 | 46.71 | 0.115 |
| barkour_vb | 53.25 | 53.51 | 55.09 | 54.98 | 0.128 |
| badger | 53.00 | 52.89 | 15.98 | 40.53 | 0.118 |
| bittle | 36.19 | 36.82 | 45.68 | 44.92 | 0.080 |
| unitree_h1 | 54.06 | 53.43 | 54.47 | 6.00 | 0.097 |
| unitree_g1 | 79.06 | 78.81 | 78.93 | 0.86 | 0.088 |
| talos | 108.37 | 108.40 | 3.25 | 10.17 | 0.088 |
| robotis_op3 | 89.31 | 88.56 | 102.63 | 98.25 | 0.103 |
| nao_v5 | 83.56 | 83.42 | 91.56 | 86.38 | 0.107 |
| cassie | 79.13 | 79.15 | 1.19 | 1.64 | 0.074 |
| hexapod | 74.94 | 76.40 | 1.28 | 0.37 | 0.075 |
| mean | 63.30 | 63.35 | 39.42 | 37.57 | 0.097 |

Figure 2: Fraction of negative pairwise gradient cosine similarities.

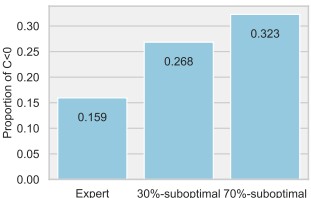

(a) Suboptimal data vs. fraction of $C < 0$.

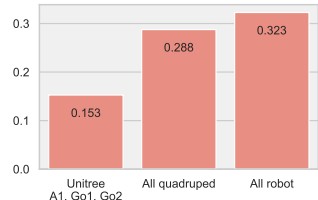

(b) Embodiment diversity vs. fraction of $C < 0$.

## 5.1 GRADIENT CONFLICTS AMONG ROBOTS

We hypothesize that when suboptimal data dominate the training set, simultaneous cross-embodiment learning induces negative transfer through conflicting policy gradients across robots. In particular, bipedal robots (e.g., Unitree H1 and Unitree G1) are most affected, since the dataset contains relatively little data from similarly embodied robots; these gradient conflicts can effectively cancel useful updates and stall learning.

To quantify this effect, we analyze the actor gradients arising from advantage-weighted regression (Kostrikov et al., 2021; Nair et al., 2020). For each embodiment $\tau$, we define the actor objective

$$\mathcal{L}_\tau^\pi(\theta) = -\mathbb{E}_{(s,a)\sim\mathcal{D}_\tau}\left[w(s,a) \log \pi_\theta(a \mid s)\right], \qquad w(s,a) = \exp\big(\beta(Q(s,a) - V(s))\big). \quad (1)$$

in which $Q(s,a)$ denotes the learned state-action value function and $V(s)$ denotes the learned state value function. We denote the per-embodiment actor gradient by $g_\tau = \nabla_\theta \mathcal{L}_\tau^\pi(\theta)$ and measure inter-embodiment alignment via the pairwise cosine similarity

$$C[\tau_i, \tau_j] = \frac{\langle g_{\tau_i}, g_{\tau_j}\rangle}{\|g_{\tau_i}\|\,\|g_{\tau_j}\|}. \quad (2)$$

Figure 2a reports, during training, the proportion of pairwise cosine similarities that are negative ($C[\tau_i, \tau_j] < 0$) for three datasets: Expert Forward and 30% and 70% Suboptimal Replay Forward. As the proportion of suboptimal data increases, the share of negative cosines increases, indicating more frequent gradient conflicts between robots.

To further investigate the relationship between transfer and gradient conflicts, we compute the correlation between each robot's transfer gain and its average gradient cosine similarity with all other robots in the 70% Suboptimal Replay dataset. We define the transfer gain as the change in average return induced by cross-embodiment training compared to single-embodiment training. We use robots with substantial transfer gain (absolute change in return greater than 10), as highlighted in Table 2. The resulting correlation, $r = 0.815$, indicates a strong positive relationship: robots that exhibit positive transfer have more aligned gradients, whereas those with large negative transfer exhibit greater gradient conflict. These findings confirm that gradient conflicts underlie the negative transfer observed when applying cross-embodiment learning to datasets rich in suboptimal data.

Next, we examine how gradient conflicts change as the number and diversity of robots increase (Figure 2b). Using the 70% Suboptimal Forward dataset, we progressively expanded the set of included

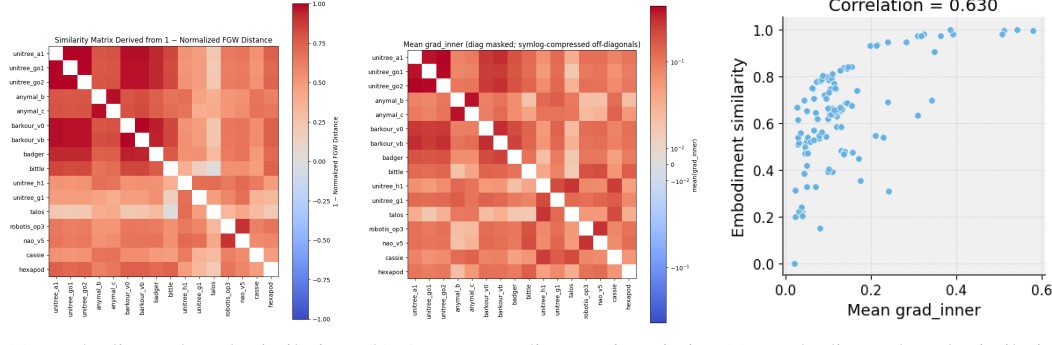

(a) Embodiment-based similarity matrix

(b) Average gradient cosine similarity matrix

(c) Embodiment-based similarity vs. mean gradient cosine similarity

Figure 3: (a) Embodiment-based similarity matrix (1 - min-max-normalized FGW distance between robot pairs); (b) Gradient cosine similarity matrix in Expert Forward dataset from Section 5.1, with the color scale compressed for readability using a symlog normalization; (c) Scatter plot of embodiment-based similarity and mean gradient cosine similarity for all robot pairs.

robots: a relatively similar group (Unitree A1, Go1, Go2), then all nine quadrupeds, and finally all 16 robots. As we include more diverse robots, each robot exhibits a higher fraction of negative pairwise cosine similarities with the others ($C < 0$). In other words, greater embodiment diversity leads to more negative cosine similarities and more frequent gradient conflicts, making negative transfer more likely. Additional distributional analysis of these effects is provided in Appendix F, and Appendix G.1 shows that the same qualitative relationships between the amount of suboptimal data, the number and diversity of embodiments, and how often gradient conflicts occur also hold for another offline RL algorithm, TD3+BC (Fujimoto & Gu, 2021), suggesting that these phenomena arise more generally in cross-embodiment offline RL.

## 5.2 CORRELATION BETWEEN EMBODIMENT DISTANCE AND THE GRADIENT CONFLICTS

We now analyze how gradient alignment across robots is captured by morphological similarity. We represent each robot's embodiment as a graph to quantify inter-robot distances. Nodes correspond to the torso, joints, and feet. Edges connect the torso to adjacent joints, adjacent joints to each other, and each terminal joint to its foot. Node features include relative positions from the torso and control parameters, capturing morphology and actuation. We then compute pairwise distances between these graphs using the Fused Gromov–Wasserstein (FGW) distance (Vayer et al., 2020). Because FGW jointly accounts for both graph structure and node features when computing distances, it is well suited for measuring dissimilarities between these robot embodiment graphs. For visualization and subsequent correlation analysis, we linearly normalize the FGW distance matrix so that all entries lie in the range $[0, 1]$, then convert distances into similarities by plotting $1 - d_{\text{FGW}}$, so that larger values correspond to more similar robot pairs. The resulting similarity matrix is shown in Figure 3a. Quadrupedal robots cluster closely, for example Unitree Go1, A1, Go2, and Anymal B/C, matches intuition.

We next analyze how these embodiment based distances relate to the gradient conflicts characterized in Section 5.1 and find that they are strongly correlated. Figure 3b presents the average gradient cosine similarity matrix from Section 5.1. Figure 3c further visualizes this relationship as a scatter plot between embodiment similarity and mean gradient cosine similarity. The Pearson correlation coefficient between these two quantities is $r = 0.63$ ($p = 1.26 \times 10^{-14}$), indicating a statistically significant positive correlation: morphologically similar robot pairs tend to have more aligned policy gradients, whereas morphologically dissimilar pairs exhibit greater gradient conflicts. Repeating the same embodiment similarity analysis with TD3+BC in Appendix G.2 yields a comparable Pearson correlation between embodiment similarity and mean gradient cosine similarity, further supporting that the morphology dependent structure of gradient conflicts is a general property. Motivated by these observations, the next subsection introduces an embodiment grouped offline RL update that explicitly exploits this structure to mitigate gradient conflicts.

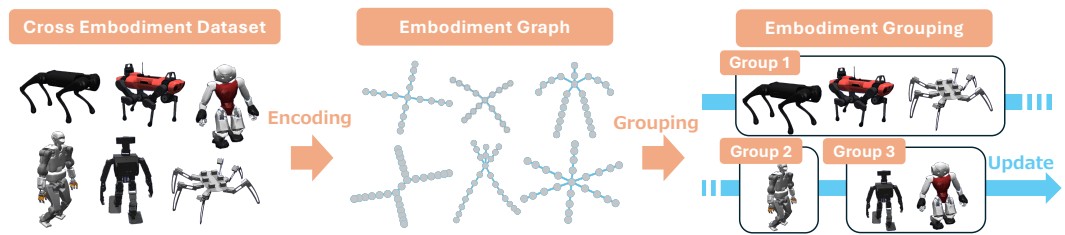

Figure 4: Overview of Embodiment Grouping (EG) for cross-embodiment offline RL.

## 5.3 EMBODIMENT-GROUPED OFFLINE RL UPDATE

Based on the analysis in the previous subsection, we hypothesized that preventing gradient conflict would yield more efficient learning. Inspired by selective group updates (Jeong & Yoon, 2025), we partition robots into groups before applying actor updates. Concretely, following Section 5.2, we represent each robot as a graph, compute pairwise FGW distances between robot graphs, and create an embodiment-distance matrix. We then apply hierarchical clustering to this matrix to obtain $M$ robot groups, which remain fixed throughout training. Training then proceeds with one global critic update, followed by sequential group-wise actor updates. Figure 4 illustrates this full

---

**Algorithm 1** Embodiment-Grouped Offline RL

**Require:** Robot groups $\{\mathcal{G}_1, \ldots, \mathcal{G}_M\}$, dataset $\mathcal{D}$
**Ensure:** Policy $\theta_\pi$; critics/targets $\theta_c$
1: **for** each outer iteration **do**
2:     Sample global minibatch $\mathcal{B} \sim \mathcal{D}$
3:     Compute $L_{\text{critic}}(\mathcal{B})$; update $\theta_c$
4:     Update target networks
5:     $P \leftarrow \text{shuffle}(\{1, \ldots, M\})$
6:     **for** $m \in P$ **do**
7:         $\mathcal{B}_m \leftarrow \{x \in \mathcal{B} \mid \text{robot}(x) \in \mathcal{G}_m\}$
8:         Compute $L_\pi(\mathcal{B}_m)$; update $\theta_\pi$
9:     **end for**
10: **end for**

---

Embodiment Grouping (EG) workflow. Details on the robot graph construction and FGW distance hyperparameters are given in Appendix E. In the experiments, we also evaluate alternative groupings such as random partitioning and a heuristic split, and we report the comparisons in Section 6.2. This pre-specified embodiment-based grouping is simple to implement yet highly effective. The algorithm is summarized in Algorithm 1 and can be easily integrated with standard offline RL methods.

## 6 EXPERIMENTAL EVALUATION

Our experiments are designed to validate the effectiveness of Embodiment Grouping in two parts. First, we evaluate performance improvements in cross-embodiment offline RL on six datasets containing varying proportions of suboptimal data. We compare eight methods. As single-method baselines, we include Behavior Cloning (BC) using the same network architecture as our offline RL backbones, TD3+BC (Fujimoto & Gu, 2021) as a strong and widely used offline continuous-control baseline, and implicit Q-learning (IQL). To evaluate existing approaches for mitigating negative transfer, we also test two gradient-conflict–aware IQL variants: IQL+PCGrad, which resolves gradient conflicts via gradient projection (Yu et al., 2020), and IQL+SEL, which selectively groups tasks based on dynamic task affinity (Jeong & Yoon, 2025). Finally, to assess the effect of our embodiment-based grouping strategy across different learning backbones, we report Embodiment Grouping (EG) counterparts of BC, TD3+BC, and IQL. We denote these variants as BC+EG, TD3+BC+EG, and IQL+EG (ours), respectively. All methods are trained offline and in aggregate use approximately 100 million transition samples per robot for gradient updates.

Second, we conduct ablation studies to isolate the contributions of (i) grouping strategy, comparing random grouping, an intuitive biped / quadruped split and our embodiment-based grouping; (ii) the number of groups $M$; and (iii) update-count control, for which we report a compute-normalized comparison that matches both total optimizer steps and processed samples across methods. Detailed hyperparameters and training configurations are provided in Appendix I.

Table 3: Each Algorithm Performance across Dataset ($\pm$ is Standard Error, 5 seeds)

| Dataset | BC | TD3+BC | IQL | IQL + SEL | IQL + PCGrad | BC + EG | TD3+BC + EG | IQL + EG (ours) |
|---|---|---|---|---|---|---|---|---|
| Expert Forward | $63.31 \pm 0.10$ | $52.14 \pm 1.89$ | $63.39 \pm 0.05$ | $63.37 \pm 0.07$ | $63.37 \pm 0.04$ | $63.47 \pm 0.04$ | $59.34 \pm 1.19$ | $\mathbf{63.52 \pm 0.04}$ |
| Expert Backward | $67.17 \pm 0.01$ | $47.94 \pm 0.48$ | $67.10 \pm 0.01$ | $\mathbf{67.24 \pm 0.02}$ | $67.05 \pm 0.02$ | $\mathbf{67.24 \pm 0.02}$ | $51.98 \pm 1.26$ | $\mathbf{67.24 \pm 0.01}$ |
| Expert Replay Forward | $49.71 \pm 1.06$ | $55.66 \pm 0.84$ | $54.61 \pm 0.12$ | $55.01 \pm 0.55$ | $53.84 \pm 0.67$ | $51.89 \pm 0.65$ | $\mathbf{57.04 \pm 0.46}$ | $54.62 \pm 0.53$ |
| Expert Replay Backward | $42.87 \pm 1.32$ | $52.31 \pm 1.22$ | $51.86 \pm 1.56$ | $55.73 \pm 1.06$ | $55.94 \pm 1.14$ | $48.64 \pm 2.65$ | $55.67 \pm 1.30$ | $\mathbf{57.58 \pm 0.05}$ |
| 70% Suboptimal Forward | $30.52 \pm 3.10$ | $35.74 \pm 1.51$ | $36.62 \pm 1.02$ | $44.59 \pm 2.02$ | $39.63 \pm 1.95$ | $42.99 \pm 1.23$ | $43.41 \pm 1.54$ | $\mathbf{51.19 \pm 1.06}$ |
| 70% Suboptimal Backward | $41.42 \pm 0.71$ | $34.79 \pm 1.40$ | $38.69 \pm 0.89$ | $44.45 \pm 1.75$ | $41.04 \pm 1.10$ | $46.30 \pm 2.39$ | $40.88 \pm 0.83$ | $\mathbf{49.60 \pm 2.39}$ |
| Mean | 49.17 | 46.43 | 52.05 | 55.07 | 53.48 | 53.42 | 51.39 | **57.29** |

## 6.1 Grouping Robots by Embodiment Makes Cross-Embodiment Learning Better

In this experiment, we evaluate, on our cross-embodiment offline RL benchmark, how much the proposed Embodiment Grouping (EG) improves implicit Q-learning (IQL) under cross-embodiment training. We compare across six datasets (Expert / Expert-Replay / 70% Suboptimal $\times$ Forward / Backward). Table 3 reports the final returns, averaged over five random seeds with $\pm$ indicating standard error. The variant that combines IQL with Embodiment Grouping achieves the best average performance. The gains are especially large when the dataset contains more suboptimal trajectories, as in the replay and 70% Suboptimal splits. Relative to methods that suppress gradient conflicts such as PCGrad and Selective Grouping (SEL), IQL+EG performs better. Compared to the IQL cross-embodiment baseline, the average improvement in the Suboptimal datasets 70% is 7.15% for PCGrad, 18.33% for SEL and 33.99% for EG.

EG yields consistent benefits across a range of offline learning baselines. On the 70% Suboptimal datasets, applying EG to TD3+BC improves performance by +19.5% on average, raising the overall mean from 46.43 to 51.39. Similarly, BC+EG improves BC by +26.3% on the same suboptimal splits. These results indicate that embodiment-aware grouping remains broadly effective across different offline objectives. These improvements are most pronounced exactly in the regimes where Section 5.1 revealed severe inter-robot gradient conflicts, supporting our hypothesis that embodiment-based grouping primarily helps by mitigating such conflicts across embodiments.

## 6.2 Ablations & Compute-Normalized Analysis

This section disentangles design choices in Embodiment Grouping (EG) and the impact of compute. We examine: (i) grouping strategy, (ii) sensitivity to the number of groups $M$, and (iii) comparisons where optimization steps and data usage are normalized.

**(i) Grouping strategy** We compare clustering based on embodiment distance (FGW; Sec.5.2) (**EG**) against random partitioning (**Random**) and an intuitive four-group split (**Heuristic**: bipeds, quadrupeds, hexapods, and torso-less bipeds). Tab.4 reports the mean final return and variance on the 70% Suboptimal Forward dataset.

From the table, **EG** achieves the most stable and substantial improvement on the 70% Suboptimal Forward dataset (+14.41,

Table 4: Ablation on grouping strategy (70% Suboptimal Forward). Values are mean final return $\pm$ SEM; Relative is vs. cross-embodiment IQL.

| Method | Final return | Relative % |
|---|---|---|
| IQL (baseline) | $37.57 \pm 0.78$ | 0.00% |
| Random grouping | $38.73 \pm 2.03$ | +3.08% |
| Heuristic | $34.45 \pm 1.97$ | -8.31% |
| EG (ours) | $\mathbf{51.98 \pm 1.70}$ | **+38.34%** |

+38.34%). In contrast, **Random** yields only a small gain (+1.16, +3.08%), and the intuitive four-way split **Heuristic** actually degrades performance (-3.14, -8.31%). This trend is consistent with the correlation between reward gains and inter-robot gradient alignment shown in Sec. 5.1, indicating that EG, which groups by FGW-based distance, better aligns updates during training. It is somewhat surprising that the Heuristic split did not improve performance. A likely reason is that coarse categories such as leg count cannot capture gradient-relevant factors like actuator placement, link lengths, mass distribution, and joint couplings. By contrast, EG based on FGW distance forms groups that reflect structural and actuation similarity while maintaining sufficient data per group, thereby preserving within-group update consistency and suppressing gradient conflicts.

**(ii) Sensitivity to the group count** $M$ We evaluate the effect of the number of Embodiment Grouping clusters $M$ by sweeping $M$ over $\{1, 2, 4, 7, 10, 13\}$. The final performance as a function of $M$ is plotted in Fig. 5; performance peaks at small to moderate $M$, whereas excessive partitioning leads to a slight degradation (e.g., on the 70% Forward dataset the best score is attained at $M{=}7$). Because $M$ also determines the number of policy updates per batch, smaller $M$ reduces the number of updates and thus improves computational efficiency. As shown in Fig. 5, the wall-clock training time grows substantially with $M$. Taken to-

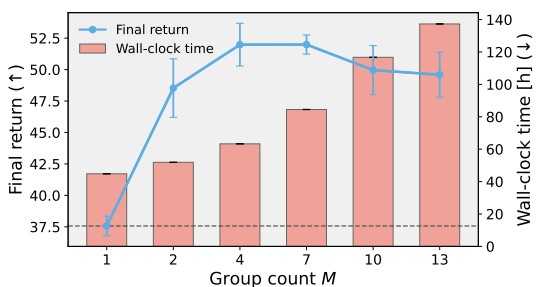

Figure 5: Effect of the number of Embodiment Grouping clusters $M$ on final return and wall-clock training time (mean $\pm$ s.e.).

gether, these results suggest that relatively small values, $M{=}2 \sim 4$, already yield strong gains; a practical strategy is to start with a small $M$ and increase it gradually while balancing the trade-off between training time and performance.

**(iii) Compute- and data-normalized comparison** EG performs $M$ policy updates per outer iteration, so a naive setup increases the number of updates by a factor of $M$. The results above matched the total number of data steps but not the number of policy's optimizer updates, which may be unfair. To remove this effect, we also run a compute-normalized comparison in which, for the IQL baseline, we multiply the total number of optimizer steps $K$ by $M$ to match EG, and reduce the batch size by $1/M$ so that the total number of processed samples remains constant. The results are sum-

Table 5: Normalized IQL vs. EG (70% Suboptimal Forward). $\Delta R$ = EG $-$ normalized IQL.

| Method | (mean $\pm$ SEM) |
|---|---|
| Normalized IQL | $44.20 \pm 2.22$ |
| IQL + EG | $\mathbf{51.98 \pm 1.70}$ |
| $\Delta R$ | $+7.78$ |

marized in Tab. 5. We observe that with the 70% Suboptimal Forward dataset, the advantage of EG persists even after normalization. This indicates that the benefit of EG does not stem merely from increasing the number of policy updates, but rather from its grouping strategy that mitigates gradient interference.

## 7 Conclusion

We presented the systematic study of combining offline RL with cross-embodiment pre-training across 16 robot platforms. Our results show that this paradigm is particularly effective when datasets contain substantial suboptimal trajectories. On such datasets it achieves higher returns than BC, and for downstream adaptation it learns faster than training without cross-embodiment pre-training. We also identified a core failure mode, inter-robot gradient conflicts, whose incidence grows with both the proportion of suboptimal data and the number of embodiments. Quantifying gradient alignment showed a consistent association with transfer performance, and we found that this alignment is well captured by embodiment distance. This motivates our proposed Embodiment Grouping (EG), which clusters robots by morphology-aware distances and performs group-wise policy updates. EG consistently reduces interference and delivers the largest gains under high-suboptimal conditions.

This study has limitations. Our evaluation is restricted to MuJoCo locomotion in simulation, which leaves sim-to-real transfer and broader domains such as manipulation and mobile manipulation untested. Our proposed Embodiment Grouping is static and derived from embodiment graphs using FGW distances. It works stably in offline RL, but it may not adapt when learning dynamics or data quality change, as in offline-to-online RL settings. Promising directions include validation on real robots and on manipulation tasks. Another direction is to improve Embodiment Grouping with dynamic grouping that uses embodiment information together with learning progress and data characteristics. In addition, once compatible groups are identified, learning group-specific embodiment or task representations within each group, for example using contrastive objectives (Yap & Koon Ng, 2024), could further strengthen within-group sharing. We leave this combined direction for future work.

ACKNOWLEDGMENTS

This work was partially supported by JST Moonshot R&D Grant Number JPMJPS2011, CREST Grant Number JPMJCR2015 and Basic Research Grant (Super AI) of Institute for AI and Beyond of the University of Tokyo. T.O. was supported by JSPS KAKENHI Grant Number JP25K03176.

REPRODUCIBILITY STATEMENT

We ensure reproducibility via clear pointers. The grouped-update procedure appears in Algorithm 1. The architecture and training details are in Appendix D; Hyperparameters for BC, IQL, TD3+BC, and PPO are in Appendix I and Tables 11 and 12. Dataset construction, including Expert, Expert Replay, and $X\%$ Suboptimal, is in Appendix B, with distributions of rewards in Appendix C. Embodiment distance and clustering, including robot graphs and FGW settings, are in Appendix E and Appendix E.2.

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

## A   USE OF LARGE LANGUAGE MODELS (LLMS)

We used LLMs to polish the writing of the paper and perform grammar checks.

# B  DATASET CONSTRUCTION DETAILS

This appendix describes how each dataset used in our experiments was constructed.

## B.1  EXPERT DATASET

The *Expert* dataset was collected using an expert model that can walk almost perfectly according to a given command. For each robot, we independently trained a PPO policy to convergence and used it to generate data. Starting from the environment's default initial state, we sampled actions from the Gaussian distribution predicted by the trained policy to log trajectories. We illustrate the *Forward* and *Backward* locomotion commands in Figure 6. For the Forward dataset, we issued a command to walk forward at $1\,\mathrm{m/s}$; for the Backward dataset, we issued a command to walk backward at $1\,\mathrm{m/s}$.

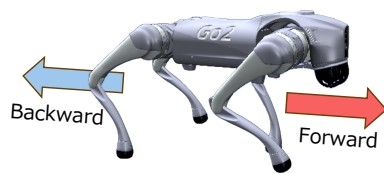

Figure 6: Illustration of the Forward and Backward locomotion commands used in our datasets.

## B.2  EXPERT REPLAY DATASET

The *Expert Replay* dataset contains interaction data collected from the beginning of training up to the point at which the model can walk according to the given command (e.g., move forward at $1\,\mathrm{m/s}$ for the *Forward* condition). Storing all PPO interaction data is impractical due to PPO's low sample efficiency, which would result in extremely large data volume and step counts. We therefore constructed a uniformly thinned $1\,\mathrm{M}$-step dataset via the following procedure:

**(1) Environment selection and full logging.**  Among the 48 parallel environments used for training, we selected one and fully recorded all rollouts (approximately $10\,\mathrm{M}$ steps) in that environment, thereby capturing trajectories from the initial exploration phase through to near convergence.

**(2) Extract data up to just before convergence.**  From the saved logs, we reconstructed episode boundaries and computed each episode's return and length. We then applied a moving average to episode returns and identified the first point at which performance reached $90\%$ of the final performance. Data up to just before this point were retained as the candidate set, while the subsequent steps, during which performance increases only slowly toward full convergence, were omitted.

**(3) Uniform thinning to $1\,\mathrm{M}$ steps.**  If the total number of steps in the candidate set exceeded $1\,\mathrm{M}$, we down-sampled by discarding episodes at equal intervals with respect to cumulative steps. This preserved the overall distribution while reducing the dataset to approximately $1\,\mathrm{M}$ steps.

Using this shared procedure, we created the replay datasets for all robots. For the *Forward* datasets of Unitree A1, Go1, and Go2, the default PPO hyperparameters led to local optima. To encourage exploration and avoid such local optima, we trained these policies with an increased policy entropy coefficient, `entropy_coef = 0.1`, and used the resulting data.

## B.3  $X\%$ SUBOPTIMAL DATASET

The $X\%$ Suboptimal dataset is constructed so that $X\%$ of the data are suboptimal. In particular, the $70\%$ *Suboptimal* dataset used in our experiments contains a relatively large proportion of suboptimal trajectories. Whereas the Replay dataset samples evenly from early to late training phases, the $X\%$ Suboptimal dataset is formed by sampling $X\%$ from the early training phase and $100-X\%$ from the late training phase.

## B.4  REWARD FUNCTION

We construct our datasets by following the walking-task setup of Bohlinger et al. (2024) and adopt a dense locomotion reward that encourages accurate tracking of commanded base velocities while penalizing undesirable gait patterns. The reward is composed of two command-tracking terms defined with respect to the target velocities $c$, together with several penalty terms that shape the gait. Each

term is weighted by its own coefficient, and the total reward is obtained by summing all weighted terms and clipping the result from below at zero. Tables 6 and 7 list the equations for each component of the reward function and the corresponding coefficients.

| Term | Name | Equation |
|------|------|----------|
| (T1) | Xy velocity tracking | $\exp\left(-|v_{xy} - c_{xy}|^2/0.25\right)$ |
| (T2) | Yaw velocity tracking | $\exp\left(-|\omega_{\text{yaw}} - c_{\text{yaw}}|^2/0.25\right)$ |
| (T3) | Z velocity penalty | $-|v_z|^2$ |
| (T4) | Pitch-roll velocity penalty | $-|\omega_{\text{pitch, roll}}|^2$ |
| (T5) | Pitch-roll position penalty | $-|\theta_{\text{pitch, roll}}|^2$ |
| (T6) | Joint nominal differences penalty | $-|q - q_{\text{nominal}}|^2$ |
| (T7) | Joint limits penalty | $-\mathbf{1}(0.9q_{\text{min}} < q < 0.9q_{\text{max}})$ |
| (T8) | Joint accelerations penalty | $-|\ddot{q}|^2$ |
| (T9) | Joint torques penalty | $-|\tau|^2$ |
| (T10) | Action rate penalty | $-|\dot{a}|^2$ |
| (T11) | Walking height penalty | $-|h - h_{\text{nominal}}|^2$ |
| (T12) | Collisions penalty | $-n_{\text{collisions}}$ |
| (T13) | Air time penalty | $-\sum_f \mathbf{1}(p_f)(p_f^T - 0.5)$ |
| (T14) | Symmetry penalty | $-\sum_f \bar{\mathbf{1}}(p_f^{\text{left}})\bar{\mathbf{1}}(p_f^{\text{right}})$ |

Table 6: Reward terms composing the reward function. Coefficients for each robot are listed in Table 7.

| Robot | T1 | T2 | T3 | T4 | T5 | T6 | T7 | T8 | T9 | T10 | T11 | T12 | T13 | T14 | T |
|-------|----|----|----|----|----|----|----|----|----|-----|-----|-----|-----|-----|---|
| ANYmalB | 2.0 | 1.0 | 2.0 | 0.05 | 0.2 | 0.0 | 10.0 | 2.5e-7 | 2e-4 | 0.01 | 30.0 | 1.0 | 0.1 | 0.5 | 20e6 |
| ANYmalC | 2.0 | 1.0 | 2.0 | 0.05 | 0.2 | 0.0 | 10.0 | 2.5e-7 | 2e-4 | 0.01 | 30.0 | 1.0 | 0.1 | 0.5 | 20e6 |
| Barkour v0 | 3.0 | 1.5 | 2.0 | 0.05 | 0.2 | 0.0 | 10.0 | 2.5e-7 | 2e-4 | 0.01 | 30.0 | 1.0 | 0.1 | 0.5 | 15e6 |
| Barkour vB | 2.0 | 1.0 | 2.0 | 0.05 | 0.2 | 0.0 | 10.0 | 2.5e-7 | 2e-4 | 0.01 | 30.0 | 1.0 | 0.1 | 0.5 | 15e6 |
| Silver Badger | 2.0 | 1.0 | 2.0 | 0.05 | 0.2 | 0.0 | 10.0 | 2.5e-7 | 2e-4 | 0.01 | 30.0 | 1.0 | 0.1 | 0.5 | 12e6 |
| Bittle | 5.0 | 2.5 | 2.0 | 0.05 | 0.2 | 0.0 | 10.0 | 2.5e-7 | 2e-4 | 0.01 | 30.0 | 1.0 | 0.1 | 0.5 | 40e6 |
| A1 | 2.0 | 1.0 | 2.0 | 0.05 | 0.2 | 0.2 | 10.0 | 2.5e-7 | 2e-5 | 0.01 | 30.0 | 1.0 | 0.1 | 0.5 | 12e6 |
| Go1 | 2.0 | 1.0 | 2.0 | 0.05 | 0.2 | 0.0 | 10.0 | 2.5e-7 | 2e-4 | 0.01 | 30.0 | 1.0 | 0.1 | 0.5 | 12e6 |
| Go2 | 2.0 | 1.0 | 2.0 | 0.05 | 0.2 | 0.0 | 10.0 | 2.5e-7 | 2e-4 | 0.01 | 30.0 | 1.0 | 0.1 | 0.5 | 12e6 |
| Cassie | 3.0 | 1.5 | 2.0 | 0.05 | 0.2 | 0.0 | 10.0 | 2.5e-7 | 2e-5 | 0.01 | 30.0 | 1.0 | 0.1 | 0.5 | 50e6 |
| Talos | 4.0 | 2.0 | 2.0 | 0.05 | 0.2 | 0.2 | 10.0 | 2.5e-7 | 2e-5 | 0.01 | 30.0 | 1.0 | 0.1 | 0.5 | 80e6 |
| OP3 | 4.0 | 2.0 | 2.0 | 0.1 | 0.2 | 0.4 | 10.0 | 1.2e-6 | 4e-4 | 6e-3 | 30.0 | 1.0 | 0.1 | 0.5 | 40e6 |
| Nao V5 | 4.0 | 2.0 | 2.0 | 0.1 | 0.2 | 0.15 | 10.0 | 1.2e-6 | 4e-4 | 6e-3 | 30.0 | 1.0 | 0.1 | 0.5 | 40e6 |
| G1 | 3.0 | 1.5 | 2.0 | 0.05 | 0.2 | 0.2 | 10.0 | 2.5e-7 | 5e-5 | 0.01 | 30.0 | 1.0 | 0.1 | 0.5 | 50e6 |
| H1 | 2.0 | 1.0 | 2.0 | 0.05 | 0.2 | 0.2 | 10.0 | 2.5e-7 | 2e-5 | 0.01 | 30.0 | 1.0 | 0.1 | 0.5 | 50e6 |
| Hexapod | 4.0 | 2.0 | 2.0 | 0.05 | 0.2 | 0.0 | 10.0 | 2.5e-7 | 2e-4 | 0.01 | 30.0 | 1.0 | 0.1 | 0.5 | 15e6 |

Table 7: Reward coefficients $r_c$ and curriculum length $T$ for each robot.

## C DATASET DETAIL

Figure 7 overlays histograms of the total reward per episode for the Forward datasets, comparing the three data quality levels used throughout the paper: Expert Forward, Expert Replay Forward, and 70% Suboptimal Forward.

Overall, three consistent patterns emerge: (i) **Expert** datasets are sharply concentrated at higher returns, indicating that most episodes achieve near-target performance; (ii) **Expert Replay** exhibits a broad spread that reflects a mixture of early failures and late competent behavior accumulated during training; (iii) **70% Suboptimal** shifts mass toward lower returns, with many episodes clustered near the low-reward region. Notably, while the 70% Suboptimal dataset contains only a small fraction of high-return episodes, those episodes tend to be considerably longer; when weighting by time steps, they account for approximately 30% of all steps. We observe qualitatively similar distributions for the Backward datasets.

These distributional differences clarify why offline RL tends to outperform pure imitation when suboptimal data are abundant: objectives that reweight actions by estimated advantages can discount

low-quality behaviors while still leveraging recoverable structure present in Replay and Suboptimal corpora.

Figure 7: Overlaid histograms of per-episode total reward (x-axis) vs. episode proportion (y-axis) for Forward datasets across all robots. Each panel corresponds to a robot; colors denote Expert, Expert replay, and 70% Suboptimal. Expert concentrates at high returns, Replay spans a wide range, and 70% Suboptimal places substantial mass on low returns. Vertical solid line: median; vertical dashed lines: 25th and 75th percentiles.

## D  ARCHITECTURE DETAILS

**Encoder (URMA).**    We split each observation into general $o_g$ and robot-specific streams. For loco-motion, the observation vector is subdivided into joint- and foot-level sets $\{o_j\}_{j \in J(\tau)}$, $\{o_f\}_{f \in F(\tau)}$ with per-item descriptors $d_j, d_f$. Descriptors and observations are encoded by MLPs $f_\phi : \mathbb{R}^{\cdot} \to \mathbb{R}^{L_d}$ and $f_\psi : \mathbb{R}^{\cdot} \to \mathbb{R}^{L_d}$. URMA uses descriptor-conditioned attention that gates each latent dimension:

$$\bar{z}_{\text{joints}} = \sum_{j \in J} z_j, \qquad\qquad z_j = \frac{\exp\left(\dfrac{f_\phi(d_j)}{\tau + \epsilon}\right)}{\sum_{L_d} \exp\left(\dfrac{f_\phi(d_j)}{\tau + \epsilon}\right)} \, f_\psi(o_j). \qquad (3)$$

In the same way, we obtain $\bar{z}_{\text{feet}}$ from $\{o_f, d_f\}$, and finally form a single latent vector by concatenation $\bar{z} = \text{concat}\big[o_g, \bar{z}_{\text{joints}}, \bar{z}_{\text{feet}}\big]$.

**Actor network.**    A core MLP $h_\theta$ maps the encoder output to an action latent:

$$\bar{z}_{\text{action}} = h_\theta(\bar{z}). \qquad (4)$$

Per joint, the action head decodes Gaussian parameters from the tuple (encoded descriptor, joint latent, action latent) and samples an action:

$$a^j \sim \mathcal{N}\big(\mu_\nu(d_j^a, \ \bar{z}_{\text{action}}, \ z_j), \ \sigma_\nu(d_j^a)\big), \qquad d_j^a = g_\omega(d_j).$$

Here $\mu_\nu$ and $\sigma_\nu$ are MLPs.

**State value network.** The state value network uses the encoder and predicts a state value from the latent:

$$V_\xi(\bar{z}) = v_\xi(\bar{z}), \tag{5}$$

implemented as an MLP (e.g., $[512, 256, 128]$).

**State–action value Network** For offline RL, we add an action encoder $f_a : \mathbb{R}^{d_a} \to \mathbb{R}^{L_a}$ and form a joint latent for $Q$:

$$z_a = f_a(a), \qquad Q_\eta^{(k)}(\bar{z}, a) = q_\eta^{(k)}\big(\text{concat}[\bar{z}, z_a]\big), \quad k \in \{1, 2\}. \tag{6}$$

To accommodate heterogeneous action sizes, we feed $f_a$ a zero-padded action vector $\tilde{a} \in \mathbb{R}^{d_{\max}}$ (padded to the maximum action length across robots).

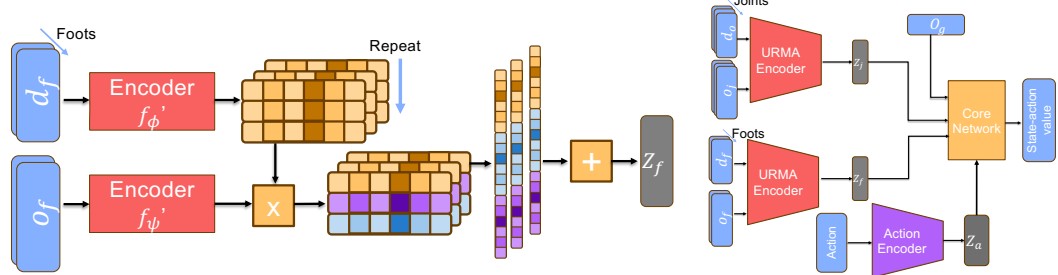

(a) URMA encoder. Descriptor-conditioned attention aggregates joint/foot latents into a fixed-size embedding.

(b) State–action value network: The action latent vector is concatenated with other latent vectors.

Figure 8: Model overview: (a) URMA encoder; (b) State-action value network.

# E   EMBODIMENT GROUPING DETAILS

This appendix gives a detailed recipe for grouping robots when building the embodiment distance used by our method. We first define the robot graph representation. We then list the node features and hyperparameters used to compute the Fused Gromov-Wasserstein distance.

## E.1   ROBOT GRAPH REPRESENTATION

We represent each robot as a graph. The nodes correspond to the torso the joints and the feet. The edges connect the torso to adjacent joints adjacent joints to each other and the terminal joint to its foot. Figure 9 shows images of all robots and the corresponding graph representation. Each node carries a feature vector constructed from the local descriptors that URMA uses as encoder inputs. The graph features are derived from the same observation factorization that URMA assumes: the joint-level and foot-level sets $\{o_j\}_{j \in J(\tau)}$ and $\{o_f\}_{f \in F(\tau)}$ with their corresponding local descriptors $\{d_j\}_{j \in J(\tau)}$ and $\{d_f\}_{f \in F(\tau)}$, together with a general observation stream. In our embodiment graphs, only the local descriptors $\{d_j\}$ and $\{d_f\}$ are used as node attributes; importantly, we do not use URMA's learned latent embeddings directly as graph node features. In particular, joint nodes use the joint descriptor $d_j$, foot nodes use the foot descriptor $d_f$, and the torso node receives a zero vector of the same dimension. The shared observation decomposition between URMA and the embodiment graphs can be viewed as an inductive bias that helps Embodiment Grouping mitigate gradient conflicts. When applying embodiment grouping with policy architectures other than URMA, it may therefore be beneficial to design graph node features so that they respect the observation factorization implied by the chosen policy architecture.

## E.2   COMPUTING FUSED GROMOV–WASSERSTEIN DISTANCE

We compute pairwise distances between robot graphs using the Fused Gromov–Wasserstein distance with the POT library (Flamary et al., 2024). Feature costs use the Euclidean norm on standardized

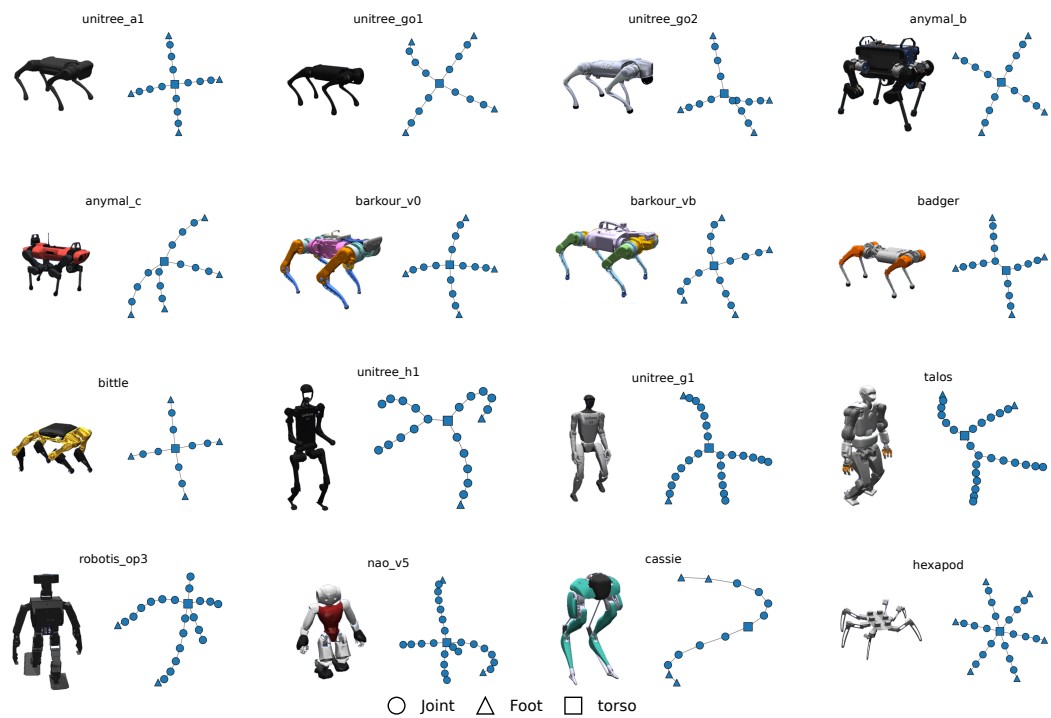

Figure 9: All robots with their images and the corresponding graph representation. Nodes are torso joints and feet. Edges follow kinematic adjacency and the connection from each terminal joint to its foot.

descriptors. Structural costs use shortest path distances on each graph. Node weights are uniform over nodes in each graph. Table 8 lists the hyperparameters that were fixed for all pairs.

We next discuss how this computation scales with the number of embodiments. In our method, FGW is used only to compute pairwise distances between embodiments before training, in order to build the embodiment distance matrix. As the number of robots $N$ increases, the number of FGW computations grows with the number of pairs, which is $\binom{N}{2} = N(N-1)/2$. Thus, the overall complexity for building the distance matrix is $O(N^2)$. In practice, this cost is small compared to policy training. For our 16-robot setup, computing the full $16 \times 16$ distance matrix with our POT-based implementation takes about 0.8 seconds on a standard machine. Even for substantially larger $N$, this one-time cost remains negligible relative to the offline RL training time.

Table 8: Essential hyperparameters for Fused Gromov–Wasserstein distance with POT

| Parameter | Setting |
| --- | --- |
| Feature ground metric | Euclidean distance |
| Structure ground metric | Shortest path distance |
| Fusion tradeoff $\alpha$ | 0.5 |
| Regularization $\varepsilon$ | 1e−3 |

### E.3 RESULTING GROUP ASSIGNMENTS

Using the FGW distance matrix described above, we perform hierarchical agglomerative clustering and cut the dendrogram to obtain $M$ groups. Figure 10 visualizes the resulting robot partitions for $M = 2, 4$. These assignments are computed once from the FGW distance matrix and kept fixed across all experiments.

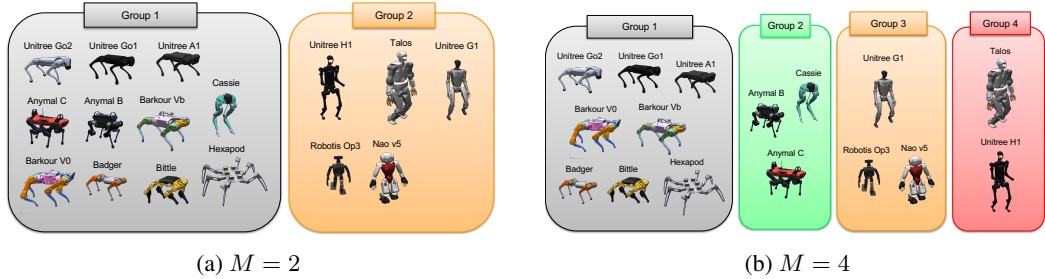

(a) $M = 2$                      (b) $M = 4$

Figure 10: Embodiment-group assignments obtained by cutting the same hierarchical clustering tree at different numbers of groups.

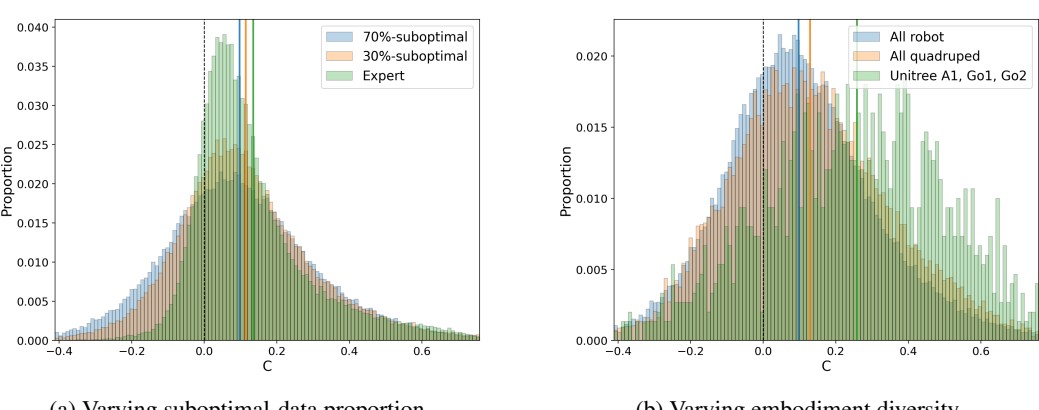

(a) Varying suboptimal-data proportion.       (b) Varying embodiment diversity.

Figure 11: **Histograms of pairwise cosine similarities** $C[\tau_i, \tau_j]$ aggregated over training. In both settings—(a) higher suboptimal-data ratios and (b) greater embodiment diversity—the fraction with $C < 0$ increases; within $C < 0$, mass concentrates at more negative values. Solid lines indicate the mean $\bar{C}$.

## F  DETAILED ANALYSIS OF GRADIENT CONFLICTS

We further analyze gradient conflicts by collecting the pairwise cosine similarities $C[\tau_i, \tau_j]$ for all robot pairs and training steps, and aggregating them into histograms (see Figure 11). Two consistent trends emerge:

**(i) Effect of suboptimal data.** As the proportion of suboptimal trajectories increases, the fraction of pairs with $C < 0$ grows. Moreover, within the negative region ($C < 0$), the share of values close to $-1$ increases with the proportion of suboptimal data, indicating a stronger misalignment per update and a higher probability of negative transfer. Consistently, the mean cosine $\bar{C}$ decreases as the suboptimal fraction increases. Overall, these results show that increasing suboptimal data makes gradient conflicts both more frequent and more severe.

**(ii) Effect of embodiment diversity.** As we include more and more diverse embodiments, the fraction of pairs with $C < 0$ increases, while the share of strongly aligned pairs (large positive $C$) diminishes. The mean cosine $\bar{C}$ also gradually declines as the embodiment diversity increases. Together, these effects indicate that greater embodiment diversity amplifies gradient conflicts, thereby increasing negative transfer and making positive transfer less likely.

## G  ADDITIONAL CORRELATION ANALYSIS FOR TD3+BC

In this appendix, we verify the generality of the IQL based analyses in Sections 5.1 and 5.2, which linked gradient conflicts to suboptimality, embodiment diversity, and embodiment similarity, by repeating the same study with another offline RL algorithm, TD3+BC. Concretely, when we apply

Table 9: Expert vs. 70% Suboptimal TD3+BC performance across robots and avg. gradient cosine similarity $C$ on the 70% suboptimal dataset. Cells shaded light red ( ) indicate *large negative transfer* (CE falls below Single by $> 10$).

| Robot | 70% Single (TD3+BC) | 70% CE (TD3+BC) | $C$ |
|---|---|---|---|
| unitree_a1 | 27.86 | 27.65 | 0.233 |
| unitree_go1 | 52.56 | 52.57 | 0.227 |
| unitree_go2 | 51.97 | 52.18 | 0.213 |
| anymal_b | 32.53 | 12.50 | 0.159 |
| anymal_c | 42.32 | 23.41 | 0.168 |
| barkour_v0 | 45.85 | 45.42 | 0.208 |
| barkour_vb | 54.78 | 54.61 | 0.235 |
| badger | 52.56 | 40.58 | 0.201 |
| bittle | 45.64 | 44.71 | 0.117 |
| unitree_h1 | 33.44 | 1.42 | 0.149 |
| unitree_g1 | 66.71 | 1.94 | 0.155 |
| talos | 73.97 | 6.14 | 0.123 |
| robotis_op3 | 101.38 | 94.35 | 0.146 |
| nao_v5 | 87.60 | 85.40 | 0.148 |
| cassie | 3.49 | 2.91 | 0.146 |
| hexapod | 23.01 | 23.24 | 0.116 |
| mean | 49.73 | 35.55 | 0.171 |

Figure 12: Fraction of negative pairwise gradient cosine similarities (TD3+BC).

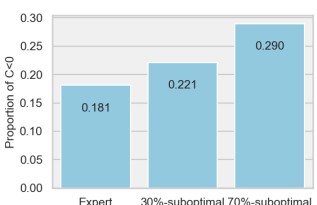

(a) Suboptimal data vs. fraction of $C < 0$ (TD3+BC).

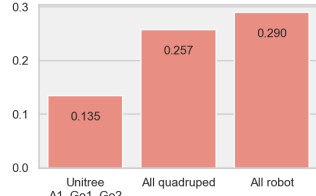

(b) Embodiment diversity vs. fraction of $C < 0$ (TD3+BC).

our gradient conflict analyses to TD3+BC, we find that (i) the fraction of negative pairwise gradient cosine similarities increases as the proportion of suboptimal data grows, (ii) the same fraction also increases as we enlarge the number and morphological diversity of robots, and (iii) these gradient conflicts are strongly structured by embodiment similarity, since robots that are closer in the morphology graph exhibit more aligned policy gradients. Taken together, the IQL and TD3+BC results indicate that the observed relationships between suboptimality, embodiment diversity, embodiment similarity, and gradient conflicts are general properties of cross embodiment offline RL rather than artifacts of a particular actor update.

### G.1 GRADIENT CONFLICTS UNDER TD3+BC.

For each embodiment $\tau$, we define the TD3+BC actor loss $\mathcal{L}_\tau^{\pi,\text{TD3+BC}}(\theta)$ on the subset of transitions belonging to robot $\tau$, and denote its gradient by $g_\tau = \nabla_\theta \mathcal{L}_\tau^{\pi,\text{TD3+BC}}(\theta)$. As in Section 5.1, we measure inter embodiment alignment using pairwise cosine similarities $C[\tau_i, \tau_j] = \langle g_{\tau_i}, g_{\tau_j} \rangle / \|g_{\tau_i}\| \|g_{\tau_j}\|$.

Figure 12a reports, over the course of training, the fraction of negative pairwise cosine similarities ($C[\tau_i, \tau_j] < 0$) under the Expert Forward, 30% Suboptimal Replay Forward, and 70% Suboptimal Replay Forward datasets. We observe the same qualitative trend as in the IQL case: as the proportion of suboptimal data increases, the fraction of negative cosines also increases, indicating more frequent gradient conflicts between robots. This confirms that the emergence of strong gradient conflicts in the presence of abundant suboptimal data is not tied to the particular choice of IQL and also occurs for TD3+BC. To further investigate the relationship between positive and negative transfer and gradient conflicts under TD3+BC, we compute the correlation coefficient between each robot's reward difference (cross-embodiment minus single-robot training) and its average TD3+BC gradient cosine similarity with all other robots in the 70% Suboptimal Replay Forward dataset, using robots with substantial performance changes (absolute reward difference greater than 10), which are highlighted in Table 9. The resulting correlation, $r = 0.766$, indicates a strong positive relationship: robots that exhibit large negative transfer exhibit greater gradient conflict. These findings mirror the IQL results and further support that gradient conflicts underlie the negative transfer observed when applying cross-embodiment learning to datasets rich in suboptimal data.

We also examine how gradient conflicts evolve with embodiment diversity under TD3+BC. Using the 70% Suboptimal Replay Forward dataset, we gradually expand the set of robots included in

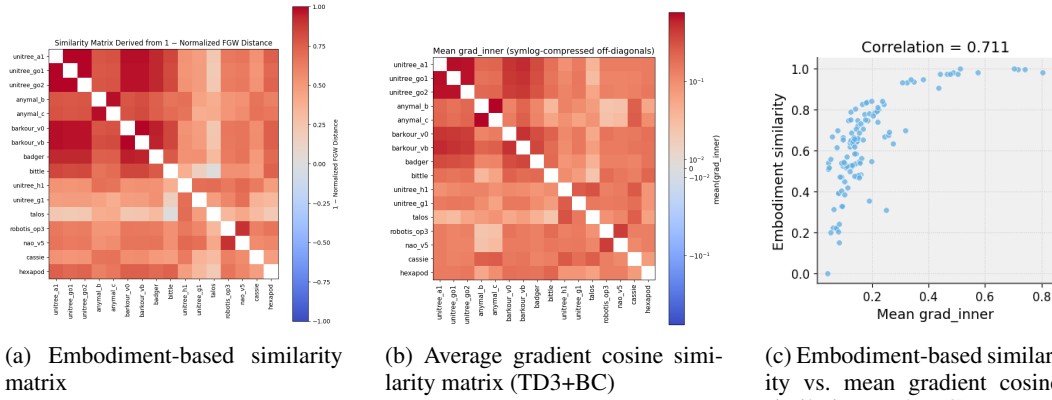

(a) Embodiment-based similarity matrix

(b) Average gradient cosine similarity matrix (TD3+BC)

(c) Embodiment-based similarity vs. mean gradient cosine similarity (TD3+BC)

Figure 13: (a) Embodiment-based similarity matrix ($1 -$ min-max-normalized FGW distance between robot pairs); (b) Average gradient cosine similarity matrix for TD3+BC on the Expert Forward dataset, with the color scale compressed for readability using a symlog normalization; (c) Scatter plot of embodiment-based similarity and mean gradient cosine similarity for all robot pairs under TD3+BC.

cross embodiment training, starting from a small, morphologically homogeneous quadruped subset (Unitree A1, Go1, Go2), then all nine quadrupeds, and finally all 16 robots. For each configuration, we compute, for every robot, the fraction of negative pairwise cosine similarities with the other robots and average this quantity across robots. Figure 12b shows that, as we add more diverse embodiments, the fraction of negative TD3+BC cosine similarities increases, just as in the IQL case. In other words, greater embodiment diversity leads to more frequent TD3+BC gradient conflicts, making negative transfer more likely when the dataset is rich in suboptimal trajectories.

### G.2 CORRELATION WITH EMBODIMENT SIMILARITY

In this section, we analyze how TD3+BC gradient alignment across robots is captured by the same embodiment similarity measure used in Section 5.2. We reuse the FGW based embodiment similarity matrix and construct the corresponding mean TD3+BC gradient cosine similarity matrix by averaging $C[\tau_i, \tau_j]$ over training for each robot pair.

Figure 13a shows the resulting embodiment-based similarity matrix, while Figure 13b presents the corresponding matrix of mean TD3+BC gradient cosine similarities. As in the IQL case, morphologically similar quadrupeds form a tight cluster in both matrices. Figure 13c further visualizes this relationship as a scatter plot between embodiment similarity and mean TD3+BC gradient cosine similarity. The Pearson correlation coefficient between these two quantities is $\mathbf{r} = \mathbf{0.711}$ with $p = 8.89 \times 10^{-20}$, indicating a strong positive correlation: robot pairs that are more similar in embodiment tend to have more aligned TD3+BC policy gradients. Together with the IQL results in Section 5.2, this supports the validity of embodiment based grouping as a general, morphology driven mechanism for reducing gradient conflicts in cross embodiment offline RL.

## H APPLYING EMBODIMENT GROUPING TO THE CRITIC

Algorithm 1 applies Embodiment Grouping (EG) to the actor updates, while the critic is updated once per outer iteration using the global minibatch. This design is motivated by our analysis that negative transfer in heterogeneous cross-embodiment offline RL is mainly caused by policy-gradient conflict. Here we examine whether extending EG to critic learning further improves performance.

**Setup.** We evaluate a variant, denoted as EG-Actor+Critic, that applies EG to the critic in addition to the actor. Specifically, after sampling a global minibatch $B$, we split it into group-specific minibatches $\{B_m\}_{m=1}^{M}$ using the same embodiment clusters as the actor update. We then update the

Table 10: Ablation on applying Embodiment Grouping to the critic. Returns are mean $\pm$ s.e. over 3 seeds on the 70% Suboptimal Replay Forward dataset.

| Group count $M$ | EG on Actor only | EG on Actor + Critic |
|---|---|---|
| 2 | $48.53 \pm 2.33$ | $48.38 \pm 2.25$ |
| 4 | $51.98 \pm 1.70$ | $50.30 \pm 0.22$ |

critic sequentially over each $B_m$ (i.e., performing $M$ critic updates per outer iteration), followed by the standard EG actor updates. All other training configurations follow Section 6.

**Results.** Table 10 reports results on the **70% Suboptimal Replay Forward** dataset. Applying EG to the critic yields no meaningful improvement over actor-only EG, and slightly degrades performance for $M = 4$.

**Discussion.** Although extending EG to the critic is conceptually possible, the results in Table 10 show that it does not bring additional return gains over the actor-only variant in our setting. Moreover, this extension comes with a clear practical drawback: the critic must be optimized separately within each group, leading to a substantial increase in wall-clock training time. EG is mainly introduced to mitigate policy-gradient conflicts, and actor-only EG already provides strong performance improvements. Given the limited benefit and the extra cost of critic-side grouping, we apply EG only to the actor in the main method.

## I  HYPERPARAMETERS AND TRAINING DETAILS

This appendix summarizes the hyperparameters and training procedures used for reproducibility. Unless otherwise noted, values are shared across all robots.

### I.1  IQL, BC, AND TD3+BC HYPERPARAMETERS FOR TRAINING

Hyperparameters for IQL, BC, and TD3+BC are listed below.

Table 11: IQL, BC, and TD3+BC hyperparameters (common)

| Parameter | IQL | BC | TD3+BC |
|---|---|---|---|
| Batch size per robot | 1024 | 1024 | 1024 |
| Learning rate | 3e-4 | 3e-4 | 3e-4 |
| Offline updates | 1e5 | 1e5 | 1e5 |
| Max grad norm | 0.5 | 0.5 | 0.5 |
| Discount factor $\gamma$ | 0.99 | – | 0.99 |
| Value expectile $\tau_{\text{exp}}$ | 0.7 | – | – |
| Policy temperature (AWAC) $\beta$ | 3.0 | – | – |
| Target network EMA $\tau_{\text{target}}$ | 0.005 | – | – |
| Policy update frequency $f_{\text{policy}}$ | – | – | 2 |
| Policy noise std $\sigma_{\text{policy}}$ | – | – | 0.2 |
| Policy noise clip $c_{\text{policy}}$ | – | – | 0.5 |
| Behavior cloning weight $\alpha$ | – | – | 2.5 |
| Max action $a_{\text{max}}$ | – | – | 5.0 |

### I.2  PPO HYPERPARAMETERS FOR DATASET GENERATION

Representative PPO hyperparameters used for dataset collection (consistent with Appendix B).

Table 12: PPO hyperparameters (dataset collection)

| Parameter | Value |
|---|---|
| Total batch size / update | 522240   (48 envs $\times$ 10880 steps) |
| Minibatch size | 32640 |
| SGD epochs per update | 10 |
| Learning rate (init $\rightarrow$ final) | 0.0004 $\rightarrow$ 0.0 (linear anneal over 100M steps) |
| Entropy coefficient | 0.0 or 0.1 |
| Discount factor $\gamma$ | 0.99 |
| GAE $\lambda$ | 0.9 |
| Clip range (PPO $\epsilon$) | 0.1 |
| Max gradient norm | 5.0 |
| Parallel environments | 48 |

