# OpenReview forum: "Cross-Embodiment Offline Reinforcement Learning for Heterogeneous Robot Datasets"
_ICLR.cc/2026/Conference — ICLR 2026 Poster_

### Official Review · Reviewer_umLy · 2025-10-28

**Soundness:** 3
**Presentation:** 2
**Contribution:** 2
**Rating:** 4
**Confidence:** 3

**Summary:**

The paper proposed cross-embodiment benchmark tasks and dataset construction paradigms for evaluating offline RL algorithms. Through empirical studies, the paper identifies both positive and negative knowledge transfer between robots with different morphology when training on suboptimal datasets. The author then hypothesized that the negative knowledge transfer happened because of morphological dissimilarity between groups of robots. Quantitatively, this can be observed from the gradient conflicts of the robot-dependent actor losses. To mitigate this issue, the paper proposed a graph-based grouping strategy using the morphology of different robots, showing better cross-embodiment offline RL performances.

**Strengths:**

- Training offline RL algorithms on cross-embodiment datasets is an important step towards scalable robotic foundation models. The paper proposed a cross-embodiment dataset collection paradigm to evaluate offline RL algorithms.

- Through empirical studies, the paper relates the negative knowledge in cross-embodiment learning to the gradient conflicts of robot-dependent actor losses.

- The paper then proposed a graph-based grouping strategy using the morphology similarity of different robots to mitigate this issue.

**Weaknesses:**

- Although the paper discussed a set of optimal and suboptimal datasets, the set of rewards / tasks are somewhat limited. Specifically, the paper only considers simple locomotion rewards (following velocities) for different robots.

- One of the motivations of the paper is to compare the performance of offline RL algorithms trained on cross-embodiment datasets. However, the paper primarily focuses on the IQL algorithm and its variants, which limit the conclusions from experiments.

- Since the paper focuses on empirical studies, including the dataset and open-source code in the submission would strengthen the conclusions.

**Questions:**

- Sec. 3.1: What’s the purpose of the reward function? Or what’s the task of each MDP? Are different robotics solving the same task?
- Sec. 3.3: The explanation of the action encoder is not clear. Do different robots share the same action support? Although they share the same action space, some action dimensions might be redundant for some robots.
- line 174: What are “Forward” and “Backward” variants of the dataset?
- Sec. 4.1: The meaning of each column in Table 1 is not clear from the text in Sec 4.1. The introductions of each baseline are deferred into Sec 6, which makes the columns in Table 1 confusing.
- Sec. 4.2 and Sec. 4.3: If I understand correctly, the single robot variant does not have a pre-training stage. Is it a fair comparison between an algorithm with cross-embodiment pre-training and task-specific fine-tuning with an algorithm only involving task-specific training? Intuitively, the number of gradient updates are different.
- The experiment results for the texts between line 311 - line 318 are missing.
- Is there a quantitative relationship or a metric between the embodiment distance (Fig 3 (a)) and the gradient cosine similarity (Fig 3 (b))?

---

> ### Author Response · Authors · 2025-11-20
> **Author Response (1/2)**
>
> We thank the reviewer for the careful reading and for highlighting the importance of cross-embodiment datasets for scalable robotic foundation models. Below, we provide responses to each of the specific points mentioned there.
>
> **1. On task and reward diversity**
> > Specifically, the paper only considers simple locomotion rewards (following velocities) for different robots.
>
> We agree that our current tasks are limited to locomotion with velocity-tracking rewards. Our main goal in this work is to analyze cross-embodiment offline RL in a setting that requires large multi-embodiment datasets with controlled suboptimality, where rewards can be defined consistently across robots, so we focus on locomotion, whose reward components and normalization are well understood. As noted in our limitations, extending the framework to richer tasks and reward structures, such as manipulation and mobile manipulation, is a natural next step once sufficiently large multi-embodiment datasets become available.
>
> **2. On the focus on IQL and the breadth of offline RL baselines**
> > However, the paper primarily focuses on the IQL algorithm and its variants,
>
> We agree that comparing multiple offline RL algorithms on cross-embodiment datasets is important. In this paper, however, our main goal is to analyze gradient conflicts and negative transfer in cross-embodiment offline RL and to evaluate the effectiveness of our embodiment-based grouping strategy. For this reason, we use IQL as a strong and widely adopted continuous-control offline RL backbone and keep the backbone fixed while varying the multi-task optimization method (PCGrad/SEL/EG).
>
> Due to limited computational resources, running a broad sweep over many algorithms across all datasets is challenging, but we have started additional experiments with other offline RL algorithms and will share these results once they are complete.
>
> **3. On dataset and code release**
> > Since the paper focuses on empirical studies, including the dataset and open-source code in the submission would strengthen the conclusions.
>
> We agree that releasing the dataset and code would strengthen the empirical contribution. Our cross-embodiment datasets are relatively large, and some parts of the data generation pipeline still need cleaning and documentation. We are in the process of preparing the code and dataset generation scripts for release, and our intention is to make them publicly available, together with trained models when possible. At submission time, we could not release the full dataset anonymously due to its size and tight coupling to institution-specific infrastructure, but if the paper is accepted we will publicly release the cleaned dataset, code, and trained models.
>
> **4. On the tasks and reward functions (Sec. 3.1)**
> > Sec. 3.1: What’s the purpose of the reward function? Or what’s the task of each MDP? Are different robotics solving the same task?
>
> Thank you for the question. In our setting, each task \tau corresponds to a robot embodiment, and all robots in a given dataset solve the same locomotion task: moving at a constant speed in a specified direction. For each robot, we train an online RL policy to track a commanded velocity and log its trajectories to build the offline dataset. The reward in each MDP uses the same components (such as xy_tracking_reward and yaw_tracking_reward), with only their weights slightly differing across robots to account for embodiment-specific properties, so the underlying task structure is shared. In the revised version, we expand Appendix B.4 with additional details on the environment and reward design, which closely follow Bohlinger’s work [1]; please refer to that work and Appendix B.4 for further information.
>
> [1] Bohlinger et al, One Policy to Run Them All: an End-to-end Learning Approach to Multi-Embodiment Locomotion, CoRL 2024
>
> **5. On the action encoder and shared action support**
> > Sec. 3.3: The explanation of the action encoder is not clear. Do different robots share the same action support?
>
> Thank you for the question. We would like to note that we described the architecture of the action encoder in Appendix D. Different robots have different action dimensions, so in the state–action value network we simply zero-pad each robot’s action to the maximum action dimension and feed the padded vector into a shared action encoder. Further implementation details of this architecture are provided in Appendix D.
>
>
> **6. On “Forward” and “Backward” dataset variants**
> > line 174: What are “Forward” and “Backward” variants of the dataset?
>
> “Forward” refers to datasets where robots are trained to walk forward, and “Backward” refers to datasets where robots are trained to walk backward, that is, to move in the opposite direction.

---

> ### Author Response · Authors · 2025-11-20
> **Author Response (2/2)**
>
> **7. On Table 1 readability and baseline introductions**
> > Sec. 4.1: The meaning of each column in Table 1 is not clear from the text in Sec 4.1. The introductions of each baseline are deferred into Sec 6, which makes the columns in Table 1 confusing.
>
> We appreciate the comment. To improve clarity, in the revised manuscript we split the original Table 1 into two tables: one associated with Section 4.1 (now Table 1) and one associated with Section 6.1 (now Table 3). This way, Table 1 in Sec. 4.1 now contains only the metrics and methods introduced in that section. Baseline comparisons that require Sec. 6 introductions are moved to Table 3 and placed next to their detailed descriptions, eliminating the need to jump between sections.
>
> **8. On pre-training versus single robot variants**
> > Is it a fair comparison between an algorithm with cross-embodiment pre-training and task-specific fine-tuning with an algorithm only involving task-specific training?
>
> We thank the reviewer for the question. Yes, the single-robot variant is trained from scratch without a pre-training stage. In Sec. 4.2, we specifically aim to measure the benefit of cross-embodiment pre-training for downstream single-robot learning, not to equalize total computation. In all plots, we match the number of gradient updates on the target robot between “single robot” and “cross-embodiment pre-trained” variants; the only difference is whether the policy is initialized from cross-embodiment pre-training or with a random initialization. This evaluation protocol follows the standard “pre-train + fine-tune vs. from-scratch” comparison widely used in vision, robot-foundation models, and multi-task RL [2,3,4].
>
> [2] He et al., Momentum Contrast for Unsupervised Visual Representation Learning, CVPR 2020.
> [3] Kim et al., Fine-Tuning Vision-Language-Action Models: Optimizing Speed and Success, RSS 2025.
> [4] Parisotto et al., Actor-Mimic: Deep Multitask and Transfer Reinforcement Learning, ICLR 2016.
>
> **9. On missing experimental results and the description around lines 311–318 in section 5.1**
> > The experiment results for the texts between line 311 - line 318 are missing.
>
> We appreciate the reviewer pointing this out. The original description was vague and did not clearly indicate which table the numbers came from, so we revised the wording for clarity. In the revised version we now write around lines 309-314 in the revised manuscript:
>
> Revised text:
>
> "*To further investigate the relationship between positive/negative transfer and gradient conflicts in cross-embodiment learning, we compute the correlation coefficient between each robot’s reward difference (cross-embodiment minus single-robot training) and its average gradient cosine similarity with all other robots in the 70% Suboptimal Replay dataset, using robots with substantial performance changes (absolute reward difference greater than $10$), which are highlighted in Table 2. The resulting correlation, $r=0.815$, indicates a strong positive relationship*"
>
> **10. On the relationship between embodiment distance and gradient cosine similarity**
> > Is there a quantitative relationship or a metric between the embodiment distance (Fig 3 (a)) and the gradient cosine similarity (Fig 3 (b))?
>
> Thank you for asking for a quantitative relationship. Following this suggestion, we computed the correlation between embodiment similarity in Fig. 3(a) and gradient cosine similarities in Fig. 3(b). The Pearson correlation coefficient is 0.63, which indicates a relatively strong relationship between morphological similarity and gradient similarity.
>
> In the revised version, we will explicitly report this Pearson correlation in Section 5.2, and we thank the reviewer for this helpful suggestion.

---

> > ### Author Response · Authors · 2025-11-26
> > **Additional results on offline RL baselines**
> >
> > **Additional results on offline RL baselines (response to 2)**
> >
> >
> > In response to the reviewer’s concern that our empirical study mainly relied on IQL, we have now completed additional experiments that combine Embodiment Grouping with another standard offline RL method. In the revised manuscript, **we add TD3+BC as an extra baseline together with its EG-augmented variant TD3+BC+EG**, and report results on all six cross-embodiment datasets. TD3+BC+EG consistently improves over TD3+BC, and on the 70% Suboptimal splits it achieves **an average relative gain of 19.5%**. Taken together with the improvements we observe for IQL and BC in Table 3, this result indicates that the benefits of **Embodiment Grouping are not limited to a particular algorithm**. EG consistently improves cross-embodiment training across value-based offline RL methods as well as behavior cloning, suggesting that grouping by morphology is a broadly effective way to mitigate negative transfer. We believe that **adding TD3+BC as an extra baseline strengthens the generality of our conclusions** while preserving the clarity of the mechanistic analysis centered on gradient conflicts. These new results are reflected in the updated Table 3 and the accompanying text in the revised version.
> >
> >
> > We observe that, on the pure Expert datasets, TD3+BC performs worse than both BC and IQL. We attribute this mainly to our use of a unified training budget and the absence of algorithm-specific hyperparameter tuning: all methods share the same number of updates, and for TD3+BC we use standard default values for key hyperparameters such as the policy update frequency and the behavior cloning weight, rather than performing any task-specific tuning. Prior work reports that offline RL methods are highly sensitive to such hyperparameters and can underperform BC when not carefully tuned, especially on near-expert data [5]. In contrast, on the Suboptimal datasets TD3+BC behaves more similarly to IQL: on the Expert Replay and 70% Suboptimal datasets it is competitive with or better than BC, and the largest gains from EG appear on the 70% Suboptimal splits. This pattern suggests that offline RL objectives are particularly important in the suboptimal regime and supports our claim that offline RL is needed for cross-embodiment learning with suboptimal demonstrations.
> >
> >
> >
> >
> > | Dataset                   | BC              | TD3+BC          | IQL             | TD3+BC+EG        | IQL+EG (ours)      |
> > |---------------------------|-----------------|-----------------|-----------------|------------------|--------------------|
> > | Expert Forward            | 63.31 ± 0.10    | 52.14 ± 1.89    | 63.39 ± 0.05    | 59.34 ± 1.19     | **63.52 ± 0.04**   |
> > | Expert Backward           | 67.17 ± 0.01    | 47.94 ± 0.48    | 67.10 ± 0.01    | 51.98 ± 1.26     | **67.24 ± 0.01**   |
> > | Expert Replay Forward     | 49.71 ± 1.06    | 55.66 ± 0.84    | 54.61 ± 0.12    | **57.04 ± 0.46**     | 54.62 ± 0.53   |
> > | Expert Replay Backward    | 42.87 ± 1.32    | 52.31 ± 1.22    | 51.86 ± 1.56    | 55.67 ± 1.30     | **57.58 ± 0.05**   |
> > | 70% Suboptimal Forward    | 30.52 ± 3.10    | 35.74 ± 1.51    | 36.62 ± 1.02    | 43.41 ± 1.54     | **51.19 ± 1.06**   |
> > | 70% Suboptimal Backward   | 41.42 ± 0.71    | 34.79 ± 1.40    | 38.69 ± 0.89    | 40.88 ± 0.83     | **49.60 ± 2.39**   |
> > | Mean                  | 49.17       | 46.43       | 52.05       | 51.39        | **57.29**          |
> >
> > [5] Kumar et al., When Should We Prefer Offline Reinforcement Learning Over Behavioral Cloning?, ICLR 2022

---

> > > ### Comment · Reviewer_umLy · 2025-11-27
> > >
> > > I thank the authors for their responses to my questions. I have three remaining points:
> > >
> > > > On the tasks and reward functions
> > >
> > > The task in multi-task RL usually refers to the same robot solving different tasks, but the multi-task RL in this paper refers to different robots solving the same task (a better phase would be multi-embodiment). I believe clarifying the difference and citing some prior work is important to understand the problem setting in Sec. 3.2. It is also important to clarify the meaning of "forward" and "backward" variants in Sec. 3.2 and mention that the rewards are simply locomotion "tasks" (also mentioned by the reviewer PQSJ).
> > >
> > > > On the relationship between embodiment distance and gradient cosine similarity
> > >
> > > For the Pearson correlation coefficient, since 0.63 is not very close to 1.0. I am curious about the scatter plot between the embodiment-based similarity score and the corresponding average gradient cosine similarity. Can we roughly see a positively correlated pattern?
> > >
> > > > Additional results on offline RL baselines
> > >
> > > The new TD3 + BC benefits from adding embodiment grouping. Does TD3 + BC also have similar patterns as shown in Figure 3 (or Table 2)? And what's the correlation coefficient for TD3 + BC?

---

> > > > ### Author Response · Authors · 2025-11-28
> > > > **Additional response to Reviewer umLy**
> > > >
> > > > We thank Reviewer umLy for the thoughtful follow-up comments and for the opportunity to further clarify our problem setting and empirical analysis. Below we address each of the three remaining points in turn.
> > > >
> > > > - **On task terminology and the multi-embodiment setting**
> > > >
> > > > We appreciate the reviewer's thoughtful comments on our use of the term “task” in our work. To clarify our problem formulation and its relation to standard multi-task RL, we have revised Sec.~3.1 as follows:
> > > >
> > > > “*We study multi-embodiment offline RL, where a single policy must control multiple robot morphologies under a common state–action interface. Unlike the standard multi-task RL setting, where a single robot embodiment solves multiple tasks with different rewards or goals, here multiple robot embodiments solve a common locomotion objective and reward functions that share the same components but may use embodiment-specific weights. Such settings have been referred to as cross-embodiment or multi-embodiment learning (Open X-Embodiment Collaboration, 2024; Bohlinger et al., 2024). In this work, we follow this terminology and refer to our setting as cross-embodiment offline RL.*”
> > > >
> > > > We have also revised Sec.~3.2 to clarify the meaning of the “Forward” and “Backward” variants and to make explicit that the underlying rewards correspond to locomotion tasks:
> > > >
> > > > “*Each of these Expert, Expert Replay, and 70\% Suboptimal Replay datasets is provided in two walking-direction variants, Forward and Backward. In the Forward variant, the robot is commanded to walk forward at $1\\mathrm{m/s}$, whereas in the Backward variant the commanded base velocity is $-1\\mathrm{m/s}$. The reward in both cases is the same dense locomotion reward, and the tasks differ only in the commanded walking direction.*”
> > > >
> > > >
> > > > Furthermore, to make the Forward and Backward variants more intuitive, we have added an illustrative figure to Figure 5 in Appendix B that visually depicts the two commanded walking directions.
> > > >
> > > > - **On the relationship between embodiment similarity and gradient cosine similarity**
> > > >
> > > > Thank you for the thoughtful follow-up question. We plotted the embodiment-based similarity score against the corresponding average gradient cosine similarity for all robot pairs, and we now include this scatter plot in the revised version (see Fig. 3(c)).
> > > >
> > > > As shown in the figure, we observe **a clear positive trend**. Robot pairs that are morphologically similar tend to exhibit larger gradient cosine similarities, whereas dissimilar pairs are concentrated around small values. This trend is summarized by the Pearson correlation coefficient of 0.63 between the two measures. When computing the Pearson correlation coefficient, we obtained a **p-value of** $\\mathbf{1.26 \\times 10^{-14}}$. This result allows us to **reject the null hypothesis of no correlation**, indicating that the correlation is statistically significant. Although the relationship is not perfectly linear, the overall increasing pattern is visually apparent and supports our use of embodiment-based grouping to reduce gradient conflicts.
> > > >
> > > > - **On extending the gradient-conflict analysis to TD3+BC**
> > > >
> > > > Thank you for this helpful suggestion. We agree that it is important to examine whether the patterns observed for IQL also appear for TD3+BC. We are currently extending our analysis to TD3+BC, computing robot-wise gradient cosine similarities and their correlation with embodiment similarity in the same way as in Fig. 3.
> > > >
> > > > These TD3+BC-specific analyses are still in progress, so we do not yet have a reliable estimate of the corresponding correlation coefficient. As soon as the experiments are complete, we will incorporate the resulting scatter plots and correlation coefficients for TD3+BC into the manuscript.

---

> > > > > ### Author Response · Authors · 2025-12-01
> > > > > **Additional response to Reviewer umLy**
> > > > >
> > > > > - **On extending the gradient-conflict analysis to TD3+BC**
> > > > >
> > > > > We thank the reviewer for encouraging us to extend our gradient conflict analysis to TD3+BC. To verify that the patterns we observed for IQL, namely that suboptimal data and greater embodiment diversity lead to stronger gradient conflicts and that embodiment similarity correlates with gradient alignment and transfer, are not specific to a single algorithm but arise more generally in cross-embodiment offline RL, we carried out the same analysis for TD3+BC. The corresponding results are included in **Appendix G.1 and Appendix G.2**.
> > > > >
> > > > > In direct response to the question about Fig. 3, we examine how embodiment similarity relates to TD3+BC gradient alignment. We reuse the FGW-based embodiment similarity measure and construct the corresponding matrix of mean TD3+BC gradient cosine similarities (Fig. 11(b,c)). The scatter plot in Fig. 11(c) shows a clear positive relationship between embodiment similarity and mean TD3+BC gradient cosine similarity, with a **Pearson correlation of $\\mathbf{r = 0.711}$ ($p = 8.89 \\times 10^{-20}$). Thus TD3+BC exhibits the same qualitative pattern as IQL: morphologically similar robots tend to have more aligned policy gradients.**
> > > > >
> > > > > In addition, we analyze how TD3+BC gradient conflicts depend on data quality and embodiment diversity. For data quality, we compute robot wise TD3+BC actor gradients and track the fraction of negative pairwise cosine similarities over training for the Expert Forward, 30\% Suboptimal Replay Forward, and 70\% Suboptimal Replay Forward datasets (Fig. 10(a)). As in the IQL case, **increasing the proportion of suboptimal data increases the fraction of negative cosine similarities, indicating more frequent gradient conflicts between robots.** For embodiment diversity, we use the 70\% Suboptimal Replay Forward dataset and gradually expand the set of robots from a small quadruped subset (Unitree A1, Go1, Go2) to all quadrupeds and then all 16 robots, and we observe that **the average fraction of negative TD3+BC cosine similarities increases as more diverse embodiments are included (Fig. 10(b))**.
> > > > >
> > > > > To relate these conflicts to transfer behavior in Table 9, we quantify each robot's performance change as the difference in its average return between cross-embodiment training and single-embodiment training. We then evaluate the correlation between this performance change and the average TD3+BC gradient cosine similarity to all other robots on the 70\% Suboptimal Replay Forward dataset. For robots with substantial performance changes (absolute reward difference greater than $10$), **the Pearson correlation is $r = 0.766$, showing that stronger negative transfer is associated with stronger gradient conflict under TD3+BC as well.** Taken together, these additional analyses address the reviewer's concerns about extending our gradient-conflict analysis to TD3+BC and support the generality of our findings beyond a single offline RL algorithm.

---

### Official Review · Reviewer_gRNY · 2025-10-29

**Soundness:** 3
**Presentation:** 2
**Contribution:** 3
**Rating:** 6
**Confidence:** 3

**Summary:**

This study investigates how to apply offline reinforcement learning (Offline RL) on cross-embodiment robotic datasets containing a large amount of suboptimal data. This study finds that when robot embodiments are diverse, gradient conflicts across different types of robots , hinder learning and lead to negative transfer. To address this issue, this study proposes an embodiment grouping (EG) strategy based on morphological similarity, which clusters morphologically similar robots into groups and updates the policy sequentially per group, effectively mitigating gradient conflicts. Experiments demonstrate that this method achieves significant performance improvements on datasets dominated by suboptimal data, with an average gain as high as 39.8%.

**Strengths:**

1.The experimental results are strong, with experiments conducted on as many as 16 different types of robots. The proposed method shows a clear improvement over the baseline.

2.The paper employs a clear validation approach to demonstrate the impact of gradient conflicts caused by cross-embodiment data.

**Weaknesses:**

1.The work lacks real-robot experiments. All experiments are conducted in simulated environments, with no validation on physical robots.

2.Certain acronym definitions appear after their first use in the paper; for example, “EG” appears in Table 1 before being formally introduced, which affects readability.

3.The implementation only evaluates forward and backward motions across different robots, lacking validation on a broader range of tasks.

**Questions:**

1.Robot locomotion can be considered to have three degrees of freedom: forward/backward, left/right, and turning left/right. This paper only addresses tasks involving the forward/backward degree of freedom. Could the authors provide results trained on all three degrees of freedom?

2.In Algorithm 1, the grouping strategy is applied only to the actor and not to the critic. What is the rationale behind this design choice?

---

> ### Author Response · Authors · 2025-11-20
> **Author Response**
>
> We thank the reviewer for the positive assessment of our contribution and the insightful feedback. Please find our response below.
>
> **1. On lack of real-robot experiments**
> >The work lacks real-robot experiments.
>
> We agree that the absence of real-robot experiments is a limitation. Our primary goal in this work is to establish and systematically analyze a cross-embodiment offline RL setting that requires large multi-embodiment datasets with precisely controlled suboptimality, which is most reliably achieved in simulation. At this scale, running the full set of experiments on physical robots is currently infeasible due to safety, cost, and time constraints. As noted in our limitations section, validating our framework and Embodiment Grouping on real robots is an important next step, and we plan to pursue this in future work.
>
> **2. On acronym definitions and presentation**
> > Certain acronym definitions appear after their first use in the paper; for example, “EG” appears in Table 1 before being formally introduced, which affects readability.
>
> We appreciate the comment on readability. To improve clarity, in the revised manuscript we split the original Table 1 into two tables: one associated with Section 4.1 (now Table 1) and one associated with Section 6.1 (now Table 3), so that “EG” no longer appears in Section 4.1 before it is introduced.
>
> **3. On task diversity and three degrees of freedom in locomotion**
> > The implementation only evaluates forward and backward motions across different robots, lacking validation on a broader range of tasks.
> >  Could the authors provide results trained on all three degrees of freedom?
>
> We agree that evaluating a broader range of locomotion behaviors is valuable. In this paper, we focus on forward and backward tasks, as our goal is to analyze a cross-embodiment offline RL setting that requires large multi-embodiment datasets with controlled suboptimality. Our dataset generation pipeline requires training an online RL policy for each robot and each commanded direction, which makes generating many such datasets computationally expensive and time-consuming.
>
> Our data generation procedure can also produce left/right and turning tasks by changing the commanded direction. We have already started constructing a dataset for lateral locomotion and running evaluations with Embodiment Grouping on this new degree of freedom, and we will share these results once they are ready.
>
> **4. On applying Embodiment Grouping only to the actor**
> > In Algorithm 1, the grouping strategy is applied only to the actor and not to the critic. What is the rationale behind this design choice?
>
> Our current design applies Embodiment Grouping to the policy because the method was specifically motivated by policy gradient conflicts. In Section 5 we analyze how policy gradients across embodiments correlate with morphological distance and show that such conflicts are a key factor in performance degradation. Given this analysis, grouping embodiments at the policy update level is a natural first step.
>
> That said, Embodiment Grouping can also be applied to the critic. One can, for example, update the critic within embodiment groups. We have started experiments that extend Embodiment Grouping to critic updates in order to study both performance effects and the impact on training time. We will share the results once they are complete.

---

> > ### Comment · Reviewer_gRNY · 2025-11-25
> > **Follow-up**
> >
> > Thank you for your detailed response.  My concerns are largely addressed. Therefore, I decided to improve my confidence of my rating.

---

> > > ### Author Response · Authors · 2025-11-26
> > > **Additional author response**
> > >
> > > We thank the reviewer for the follow up and for revisiting our submission. Below we provide additional results and analysis to further support our conclusions.
> > >
> > > - **Additional response to "3. On task diversity and three degrees of freedom in locomotion"**
> > >
> > > Following the reviewer’s suggestion and as noted in our original response, we have now constructed a dataset for leftward locomotion and run additional evaluations of Embodiment Grouping on this lateral degree of freedom. We report results for BC, IQL, and IQL+EG on 70% Suboptimal Leftward Replay and Expert Leftward Replay:
> > >
> > > | Dataset                         | BC              | IQL             | IQL+EG (ours)      |
> > > |---------------------------------|-----------------|-----------------|--------------------|
> > > | 70% Suboptimal Leftward Replay | 52.70 ± 2.37    | 51.68 ± 2.95    | **60.93 ± 5.36**   |
> > > | Expert Leftward Replay         | 59.04 ± 3.09    | 61.60 ± 0.54    | **61.94 ± 2.98**   |
> > >
> > > On the 70% Suboptimal Leftward Replay dataset, **IQL+EG improves over both BC and vanilla IQL by a large margin**, and it also slightly improves over the baselines on Expert Leftward Replay. These additional results support that Embodiment Grouping continues to provide benefits when extending from forward and backward locomotion to a lateral degree of freedom. We are further extending these experiments to additional expert datasets and offline RL algorithms, and we will report the full set of results once they are available.
> > >
> > > - **Additional response to "4. On applying Embodiment Grouping only to the actor"**
> > >
> > > Following your suggestion, we conducted an additional ablation that applies Embodiment Grouping (EG) to the critic as well.
> > > On the 70% Suboptimal Replay Forward dataset, the results are:
> > >
> > > | Group count M | EG on Actor only | EG on Actor + Critic |
> > > |---:|---:|---:|
> > > | 2 | 48.53 ± 2.33 | 48.38 ± 2.25 |
> > > | 4 | 51.98 ± 1.70 | 50.30 ± 0.22 |
> > >
> > > Overall, applying EG to the critic does not provide additional return gains over the actor-only EG in our setting. Moreover, it introduces a clear practical cost: the critic must be optimized separately for each group, which substantially increases wall-clock training time. EG mainly targets policy-gradient conflicts, and actor-only EG already yields strong gains. Therefore, we use actor-only EG as our main method, since critic-side grouping adds extra cost with limited benefit.
> > >
> > > We have added these results and the corresponding discussion in Appendix G of the revised manuscript, and we appreciate your suggestion, which helped us to improve the paper.

---

> ### Comment · Reviewer_gRNY · 2025-11-26
>
> Thanks for the author for the additional experiments. I believe your novel contribution, but it would be better if the authors consider to conduct real-world robot experiments (even if small-scale and tentative) to justify the practical significance of this work in the future. Therefore, I tend to  (weakly) accept this paper.

---

> > ### Author Response · Authors · 2025-11-27
> > **Response to Reviewer gRNY's comment**
> >
> > Thank you again for carefully reading our rebuttal and for maintaining a positive assessment of our work. In this paper, we focused on foundational aspects of cross-embodiment offline RL in a controlled setting, so real-robot experiments were out of scope this time. We agree on the importance of real-world validation and plan to explore physical robot experiments as an important direction for future work.

---

### Official Review · Reviewer_PZLb · 2025-10-31

**Soundness:** 3
**Presentation:** 3
**Contribution:** 3
**Rating:** 6
**Confidence:** 3

**Summary:**

The paper studies using cross-embodiment with offline RL on locomotion tasks. It finds that offline RL is more valuable when offline dataset is more sub-optimal. Furthermore, it notices that cross-embodiment transfer of some robots have negative impacts, owing to large morpholy gap and conflict gradients. The paper then proposes valuable method to group different groups to prevent conflict gradients. Experiment show that the proposed method has strong improvement over baseline offline RL, comparing to other multi-task gradient techniques.

**Strengths:**

1. The setting of cross-embodiment offline RL is novel in the community.
2. The proposed method is easy to understand and is effective in practice.
3. The comparison involve gradient projection techniques used in continual learning / multi-task literature,
3. The evaluation of different embodiment is comprehensive.

**Weaknesses:**

1. Tasks only contain locomotion on walking. A comprehensive analysis / benchmark should involve more diverse tasks.
2. The baseline offline RL only contains IQL. More algorithms like CQL e.t.c should be evaluated. It is unclear if the claim can be extended.

**Questions:**

1. Can the results generalize to other imitaiton learning / offline RL algorithms.

---

> ### Author Response · Authors · 2025-11-20
> **Author Response**
>
> Thank you very much for the constructive and encouraging feedback. We address the reviewer’s questions and comments below.
>
> **1. On task diversity**
> > Tasks only contain locomotion on walking. A comprehensive analysis / benchmark should involve more diverse tasks.
>
> We agree that our current benchmark focuses on locomotion and that extending to more diverse tasks is an important direction. We would like to note that extending the framework and Embodiment Grouping to broader domains such as manipulation and mobile manipulation is a natural next step. Our goal is to study the effects of scaling robot datasets across diverse morphologies in the presence of substantial suboptimality. In particular, we examine the benefits of cross-embodiment sharing, the failure modes that emerge (such as gradient conflicts), and strategies to mitigate them. To analyze this cleanly, we focus on locomotion, where body morphology has a direct and pronounced impact on behavior and is easier to interpret than in navigation or manipulation tasks, and where rewards can be consistently defined across embodiments.
>
> **2. On offline RL baselines and generalization to other algorithms**
> > The baseline offline RL only contains IQL.
>
> > Can the results generalize to other imitation learning / offline RL algorithms.
>
> We appreciate the reviewer’s suggestion to evaluate additional offline RL and imitation learning algorithms. Embodiment Grouping is a general mechanism that mitigates gradient conflicts at the level of policy updates, so it can be combined with other imitation learning and offline RL methods, not only IQL. In our experiments, we confirm that Embodiment Grouping improves performance not only for IQL but also for behavioral cloning. On the 70% Suboptimal datasets, it yields average gains of 33.99% over the IQL baseline and 26.32% over the behavioral cloning baseline.
>
> Due to limited computational resources, it is difficult to run a broad comparison over many offline RL algorithms. However, we have started additional experiments that combine Embodiment Grouping with other standard offline RL algorithms, and we will share the results once they are complete.

---

> > ### Author Response · Authors · 2025-11-26
> > **Additional results on offline RL baselines**
> >
> > **Additional results on offline RL baselines (response to 2)**
> >
> > Following our rebuttal, we have now completed the additional experiments combining Embodiment Grouping with other offline RL algorithms. In the revised manuscript, **we add TD3+BC as an extra baseline together with its EG-augmented variant TD3+BC+EG**, and report results on all six cross-embodiment datasets. TD3+BC+EG consistently improves over TD3+BC, and on the 70% Suboptimal splits it achieves **an average relative gain of 19.5%**. Taken together with the improvements we observe for IQL and BC in Table 3, this result indicates that the benefits of **Embodiment Grouping are not limited to a particular algorithm**. EG consistently improves cross-embodiment training across value-based offline RL methods as well as behavior cloning, suggesting that grouping by morphology is a broadly effective way to mitigate negative transfer. We believe that **adding TD3+BC as an extra baseline strengthens the generality of our conclusions** while preserving the clarity of the mechanistic analysis centered on gradient conflicts. These new results are reflected in the updated Table 3 and the accompanying text in the revised version.
> >
> >
> > We observe that, on the pure Expert datasets, TD3+BC performs worse than both BC and IQL. We attribute this mainly to our use of a unified training budget and the absence of algorithm-specific hyperparameter tuning: all methods share the same number of updates, and for TD3+BC we use standard default values for key hyperparameters such as the policy update frequency and the behavior cloning weight, rather than performing any task-specific tuning. Prior work reports that offline RL methods are highly sensitive to such hyperparameters and can underperform BC when not carefully tuned, especially on near-expert data [1]. In contrast, on the Suboptimal datasets TD3+BC behaves more similarly to IQL: on the Expert Replay and 70% Suboptimal datasets it is competitive with or better than BC, and the largest gains from EG appear on the 70% Suboptimal splits. This pattern suggests that offline RL objectives are particularly important in the suboptimal regime and supports our claim that offline RL is needed for cross-embodiment learning with suboptimal demonstrations.
> >
> >
> >
> >
> > | Dataset                   | BC              | TD3+BC          | IQL             | TD3+BC+EG        | IQL+EG (ours)      |
> > |---------------------------|-----------------|-----------------|-----------------|------------------|--------------------|
> > | Expert Forward            | 63.31 ± 0.10    | 52.14 ± 1.89    | 63.39 ± 0.05    | 59.34 ± 1.19     | **63.52 ± 0.04**   |
> > | Expert Backward           | 67.17 ± 0.01    | 47.94 ± 0.48    | 67.10 ± 0.01    | 51.98 ± 1.26     | **67.24 ± 0.01**   |
> > | Expert Replay Forward     | 49.71 ± 1.06    | 55.66 ± 0.84    | 54.61 ± 0.12    |**57.04 ± 0.46**     | 54.62 ± 0.53   |
> > | Expert Replay Backward    | 42.87 ± 1.32    | 52.31 ± 1.22    | 51.86 ± 1.56    | 55.67 ± 1.30     | **57.58 ± 0.05**   |
> > | 70% Suboptimal Forward    | 30.52 ± 3.10    | 35.74 ± 1.51    | 36.62 ± 1.02    | 43.41 ± 1.54     | **51.19 ± 1.06**   |
> > | 70% Suboptimal Backward   | 41.42 ± 0.71    | 34.79 ± 1.40    | 38.69 ± 0.89    | 40.88 ± 0.83     | **49.60 ± 2.39**   |
> > | Mean                  | 49.17       | 46.43       | 52.05       | 51.39        | **57.29**          |
> >
> > [1] Kumar et al., When Should We Prefer Offline Reinforcement Learning Over Behavioral Cloning?, ICLR 2022

---

> > > ### Author Response · Authors · 2025-12-01
> > > **Additional results on offline RL baselines**
> > >
> > > - **Additional analysis on gradient conflicts under TD3+BC (response to 2)**
> > >
> > > We thank the reviewer again for raising the question of whether our findings extend beyond IQL. In addition to the TD3+BC results demonstrating the effect of Embodiment Grouping reported above, we have now carried out the same gradient conflict and embodiment similarity analyses for TD3+BC to check whether the same mechanisms also appear for another standard offline RL algorithm. The new results are included in Appendix G.1 and Appendix G.2 of the revised manuscript.
> > >
> > > We first analyze how TD3+BC gradient conflicts depend on data quality, embodiment diversity, and transfer behavior. Using robot wise TD3+BC actor gradients, we track the fraction of negative pairwise cosine similarities during training on the Expert Forward, 30\% Suboptimal Replay Forward, and 70\% Suboptimal Replay Forward datasets (Fig. 10(a)). **As in the IQL case, increasing the proportion of suboptimal data increases the fraction of negative cosine similarities, indicating more frequent gradient conflicts between robots.** For embodiment diversity, we use the 70\% Suboptimal Replay Forward dataset and gradually expand the set of robots used for cross embodiment training from a small quadruped subset (Unitree A1, Go1, Go2) to all quadrupeds and finally all 16 robots. As shown in Fig. 10(b), the average fraction of negative TD3+BC cosine similarities increases as more diverse embodiments are included, meaning that **greater embodiment diversity leads to more frequent gradient conflicts under TD3+BC as well.**
> > >
> > > To relate these conflicts to transfer behavior summarized in Table 9, we quantify each robot's performance change as the difference in its average return between cross-embodiment training and single-robot training. On the 70\% Suboptimal Replay Forward dataset, we then compute, for robots with substantial performance changes (absolute reward difference greater than $10$), the Pearson correlation between this reward difference and the average TD3+BC gradient cosine similarity to all other robots. The resulting **correlation is $r = 0.766$, showing that larger negative transfer is associated with stronger gradient conflict.**
> > >
> > >
> > > We also examine how embodiment similarity relates to TD3+BC gradient alignment. Reusing the FGW-based embodiment similarity measure, we construct the corresponding matrix of mean TD3+BC gradient cosine similarities (Fig. 11(b,c)). The scatter plot in Fig. 11(c) shows a strong positive relationship between embodiment similarity and mean TD3+BC gradient cosine similarity, with a **Pearson correlation of** $\\mathbf{r = 0.711}$ ($p = 8.89 \times 10^{-20}$), indicating that **morphologically similar robots tend to have more aligned TD3+BC policy gradients.** Taken together with the IQL results, these TD3+BC findings support our claim that gradient conflicts and their dependence on embodiment similarity and data quality are general features of cross-embodiment offline RL rather than artifacts of a single algorithm.

---

### Official Review · Reviewer_PQSJ · 2025-11-01

**Soundness:** 2
**Presentation:** 3
**Contribution:** 3
**Rating:** 6
**Confidence:** 3

**Summary:**

This paper presents Embodiment Grouping (EG), a morphology-aware approach for cross-embodiment offline RL. The authors observe that gradient conflicts arise when training shared policies across diverse morphologies. They measure embodiment similarity via the Fused Gromov–Wasserstein (FGW) distance, which jointly considers kinematic structure and behavior embeddings, and use this to cluster robots into morphology-aligned groups for group-wise policy training. The result: better policy transfer, stability, and data efficiency across heterogeneous robot datasets.

**Strengths:**

1.	The paper fills an important gap between robot foundation models and morphology-aware generalization. Prior work (e.g., URMA) has explored transfer across morphologies but typically ignores the destructive gradient interference that arises when training across incompatible embodiments. EG’s use of FGW distance to quantify morphology-level relationships and structure training groups is an elegant and novel solution.

2.	The experimental setup is extensive and thoughtfully designed. Using 16 diverse morphologies (quadrupeds, bipeds, manipulators) provides a broad basis for claims. The analysis linking gradient cosine similarity and policy degradation offers convincing causal evidence. Ablation studies on group count and morphological embedding confirm robustness of results.

3.	The paper explains why shared gradient updates can harm policies across dissimilar bodies, which is a phenomenon often observed but seldom quantified. Their visualizations of inter-group gradient alignments are insightful and elevate the reader’s understanding of multi-embodiment interference.

**Weaknesses:**

1.	While FGW distance effectively captures morphological similarity, there is no formal argument linking FGW similarity to gradient alignment or loss landscape smoothness. Without such a bridge, the theoretical contribution remains descriptive rather than predictive.

2.	Embodiment relationships evolve as policies adapt. Fixed grouping may lead to stale partitions that no longer reflect actual learning dynamics.

3.	The morphological embeddings used to compute FGW distance come from URMA, which itself encodes behavioral information. This could inflate grouping performance.

4.	The experiments focus exclusively on locomotion. It’s unclear if EG transfers to manipulation tasks, where control topology differs drastically.

**Questions:**

1.	How does EG perform as the number of embodiments scales? Does the FGW computation remain tractable?

2.	Could group-wise contrastive representation learning replace fixed clustering?

3.	Is there an optimal number of groups, or does performance plateau?

---

> ### Author Response · Authors · 2025-11-20
> **Author Response (1/2)**
>
> Thank you very much for the highly constructive feedback. We provide several responses below.
>
> **1. Link between FGW similarity and gradient alignment**
> >While FGW distance effectively captures morphological similarity, there is no formal argument linking FGW similarity to gradient alignment or loss landscape smoothness.
>
> Thank you for pointing out this aspect. In Section 5.2, we empirically analyze the relationship between FGW-based embodiment similarity and gradient conflicts, and show that robots that are closer in FGW distance tend to exhibit more aligned actor gradients. Consistent with this, grouping robots based on FGW distance reduces gradient conflicts and improves performance compared to morphology-agnostic baselines.
>
> In the revised version, we additionally report the Pearson correlation between the embodiment similarity (FGW-based) in Fig. 3(a) and gradient cosine similarities in Fig. 3(b), which is $r=0.63$, indicating a relatively strong correspondence.
>
> **2. Evolving embodiment relationships and fixed grouping**
> > Fixed grouping may lead to stale partitions that no longer reflect actual learning dynamics.
>
> Thank you for this insightful question. We agree that the embodiment relationships may evolve as policies adapt and that fixed grouping could in principle become stale. In this work we focus on a static grouping derived from embodiment structure, but we explicitly compare against a dynamic grouping baseline. In particular, we include SEL, which recomputes task groups during training based only on their loss relationships. In the 70% suboptimal setting, this dynamic grouping does not outperform our embodiment-based fixed grouping and sometimes performs worse. However, as noted in the future work discussion in the Conclusion section, there are settings where relationships and data quality can shift over time, such as offline-to-online RL. In such cases, embodiment-aware dynamic grouping is a promising next step for our method.
>
> **3. URMA-based morphological embeddings and possible inflation of performance**
> > The morphological embeddings used to compute FGW distance come from URMA, which itself encodes behavioral information. This could inflate grouping performance.
>
> Thank you for raising this point. We would like to clarify that we do not use URMA’s learned latent embeddings directly as input to FGW. When constructing the embodiment graphs, each robot is represented as a graph whose nodes correspond to joints, feet, and the torso, and each node is assigned simple descriptors such as the joint/foot’s relative position to the torso. These node features follow the same observation decomposition as in URMA: joint-level and foot-level sets $\\{o_j\\}&#95;{j \in J(\tau)}$ and $\\{o_f\\}&#95;{f \in F(\tau)}$, their corresponding local descriptors $\{d_j\}&#95;{j \in J(\tau)}$ and $\{d_f\}&#95;{f \in F(\tau)}$, and a general observation. In our embodiment graphs, we use these descriptors $\{d_j\}, \{d_f\}$ as node features.
>
> In other words, the graph features are derived from the observation structure that URMA also uses, not from URMA’s trained network outputs. This shared decomposition may indeed provide an inductive bias that helps EG mitigate gradient conflicts. When applying embodiment grouping with network architectures other than URMA, it may be beneficial to design the graph node features so that they respect the observation factorization implied by the chosen policy architecture.
>
> To improve clarity, we have added the above description to Appendix E.1 in the revised manuscript.
>
> **4. Locomotion only experiments**
> > The experiments focus exclusively on locomotion.
>
> We agree that our current benchmark focuses on locomotion (walking) and that extending to more diverse tasks is an important direction. Our goal is to study the effects of scaling robot datasets across diverse morphologies in the presence of substantial suboptimality. In particular, we examine the benefits of cross-embodiment sharing, the failure modes that emerge (such as gradient conflicts), and strategies to mitigate them. To analyze this cleanly, we focus on locomotion, where body morphology has a direct and pronounced impact on behavior and is easier to interpret than in navigation or manipulation tasks, and where rewards can be consistently defined across embodiments. As noted in our limitations and future work, extending the framework and Embodiment Grouping to broader domains such as manipulation and mobile manipulation with perception is a natural next step.

---

> > ### Author Response · Authors · 2025-11-20
> > **Author Response (2/2)**
> >
> > **5. Scalability of FGW as the number of embodiments grows**
> > > How does EG perform as the number of embodiments scales? Does the FGW computation remain tractable?
> >
> > We appreciate the question about how EG scales with the number of embodiments. In our method, FGW is used only to compute pairwise distances between embodiments before training, in order to build the embodiment distance matrix. The computational cost per FGW computation for one pair of embodiments does not depend on the total number of embodiments and what grows with $N$ is simply the number of pairs, which is $\binom{N}{2} = N(N-1)/2$. Thus, the overall complexity for building the distance matrix is $O(N^2)$ in the number of robots $N$.
> >
> > In practice this cost is small compared to policy training. For our 16-robot setup, computing the full 16×16 distance matrix with our POT-based implementation takes about 0.8 seconds on a standard machine. Even for substantially larger $N$, this one-time cost remains negligible relative to offline RL training time.
> >
> > In response to this helpful question, we have added the above complexity discussion to Appendix E.2 in the revised manuscript.
> >
> > **6. Group-wise contrastive representation learning for clustering robots**
> > > Could group-wise contrastive representation learning replace fixed clustering?
> >
> > Thank you for raising this possibility. We’re not sure which specific work you had in mind, but we assume it refers to approaches such as “Group-wise Contrastive Bottleneck for Weakly-Supervised Visual Representation Learning” (Yap et al., 2024) [1]. Such methods are developed for settings where the group structure is given a priori, for example semantic attribute groups or hierarchical labels. The group-wise contrastive objective then learns representations conditioned on that fixed grouping. These methods are not intended to discover or update the grouping itself.
> >
> > By contrast, our Embodiment Grouping tackles a different setting. Its main goal is to discover how to partition embodiments into groups when the group structure is not known in advance. This clustering step is therefore central to our contribution, so group-wise contrastive learning, which assumes predefined groups, cannot replace it. Following the reviewer’s suggestion, we see group-wise contrastive learning as a promising complement to Embodiment Grouping. Once EG forms compatible groups, we could apply contrastive learning within each group to learn richer embodiment or task representations. In the revised version, we have included this as an interesting direction for future work.
> >
> > [1] Yap et al. Group-wise Contrastive Bottleneck for Weakly-Supervised Visual Representation Learning, WACV2024.
> >
> > **7. Number of groups and performance plateau**
> > > Is there an optimal number of groups, or does performance plateau?
> >
> > Thank you for asking about the optimal number of groups. We would like to note that the impact of the number of groups within our framework is discussed in Section 6.2. In particular, in Sec. 6.2(ii), we study the effect of the number of clusters M by varying M over {1,2,4,7,10,13}. The final performance peaks at small to moderate $M$, whereas excessive partitioning leads to a slight degradation (e.g., on the 70\% Forward dataset the best score is attained at $M{=}4$-$7$). Because $M$ also determines the number of policy updates per batch, smaller $M$ reduces the number of updates and thus improves computational efficiency. In practice, our results suggest that relatively small values such as $M{=}2$-$4$ already yield large gains, so a reasonable strategy is to start from a small M and increase it gradually while trading off performance against training cost.

---

> > > ### Comment · Reviewer_PQSJ · 2025-11-27
> > >
> > > Thank you for your responses. I maintain my rating as 6.

---

> > > > ### Author Response · Authors · 2025-11-27
> > > > **Response to Reviewer PQSJ's comment**
> > > >
> > > > Thank you again for your thoughtful review and for maintaining a positive rating. We sincerely appreciate the time and care you devoted to reading our work and giving detailed feedback.

---

### Author Response · Authors · 2025-12-03
**Summary of Reviewer Discussion**

## Summary of Reviewer Discussion

We would like to thank all the reviewers for their thoughtful and comprehensive reviews. In this official comment, we summarize the discussion so far.

All reviewers recognize the **novelty and importance of cross embodiment offline RL** and see our setting as an important step toward scalable robotic foundation models. They highlight that

- the paper presents a **concrete benchmark** and dataset construction paradigm, together with **extensive experiments** on sixteen diverse robot embodiments that provide strong empirical support for this setting
- the work provides a **clear and systematic analysis of gradient conflicts and knowledge transfer**, quantitatively linking morphology, suboptimality, gradient cosine similarity, and knowledge transfer across robots
- the proposed **Embodiment Grouping (EG)** is a simple, morphology aware method based on FGW and graphs that is easy to understand and consistently improves over strong offline RL and multi task gradient baselines

Reviewers also raised several key points that could be clarified or strengthened. We addressed these through additional experiments, analyses, and revisions, and we summarize the main points below.

|#|Topic|Reviewers (resp. IDs)|Reviewers' comments|Our response|
|-|-|-|-|-|
|1| IQL only and generality across algorithms | PZLb (2, follow-up), umLy (2, follow-up) | Results might depend on using only IQL as the offline RL backbone. | We added TD3+BC baselines with EG augmented variants. EG consistently improves BC, IQL, and TD3+BC, and we extended the gradient conflict analysis to TD3+BC, showing that both the empirical gains of EG and the underlying mechanisms generalize across algorithms in the cross embodiment offline RL setting. |
|2| Link between embodiment similarity and gradient conflicts | PQSJ (1), PZLb (2, follow-up), umLy (10, follow-up) | Asked for more explicit quantitative evidence connecting FGW-based embodiment similarity and gradient alignment. | We now explicitly report Pearson correlations and scatter plots for IQL, and we additionally perform the same analysis and add new plots for TD3+BC, showing a positive relationship between embodiment similarity and mean gradient cosine similarity for both methods. These results provide quantitative and visual evidence that morphology based similarity is closely tied to gradient alignment.|
|3| Design of grouping (scalability, group size, static vs dynamic, actor vs critic) | PQSJ (2, 5, 6, 7), gRNY (4) | Questions about FGW scalability, the choice of group count, fixed versus dynamic grouping, and applying EG only to the actor. | We clarified that the FGW-based similarity matrix is computed once before training and its cost is negligible compared to policy updates. The paper already analyzes how performance varies with the number of groups. We compared EG to a dynamic grouping baseline (SEL), which does not outperform EG on the suboptimal datasets, and added an ablation that applies EG to the critic, which brought little extra benefit while increasing wall clock cost.|
|4| Problem setting and task/reward clarity | gRNY (2), umLy (4, 6, 7, 9) | Requested clearer explanations of the task setting and reward function, the meaning of Forward and Backward variants, and some table and notation issues. | We rewrote Section 3 to more clearly define the cross embodiment offline RL setting, tasks, and reward, and added a figure illustrating the Forward and Backward variants. We also split the original Table 1 into two tables to improve readability.|
|5| Task diversity | PQSJ (4), PZLb (1), gRNY (1, 3), umLy (1)| Tasks are limited to simulated locomotion.| We clarified that we focus on simulated locomotion because it enables large cross embodiment datasets with controlled suboptimality and because morphology has a direct effect on behavior. To address task diversity, we added experiments on a leftward locomotion dataset where EG again improves over IQL, and we identify richer task families and real robot validation as important directions for future work.|
|6| Dataset/code release and implementation details| PQSJ (3), umLy (3, 5, 8) | Requests for dataset and code release, a clearer description of the action encoder and pretraining protocol, and clarification of how URMA related features are used for the graph node representation.| We committed to releasing cleaned datasets and scripts upon acceptance. We clarified the shared action encoder design and the rationale of the pretraining versus single robot comparison, and clarified that the graph node features reuse URMA style observation decomposition but not URMA's learned latent embeddings.|

### Overall conclusion
Overall, the reviewers’ assessments are positive, and we believe that the additional experiments and clarifications in the revised version comprehensively address the remaining points that were raised. We hope our findings will contribute to the future of scalable robotic foundation models.

---

### Author Response · Authors · 2025-12-04
**Official Comment by Authors to Area Chair**

Dear Area Chair,

We sincerely appreciate your time and effort in evaluating our work in this exceptional situation. During the discussion and rebuttal period, the four reviewers raised many helpful comments and questions, and we have responded to all of them and revised the paper accordingly. We believe that the new experiments, analyses, and clarifications comprehensively address the points raised by the reviewers and further strengthen the contribution.

For convenience, we provide below a summary of the discussion and our revisions under the heading “Summary of Reviewer Discussion.”

We are very grateful for your time and consideration.

Sincerely,
The Authors

---

### Meta-Review · Area_Chair_jW5q · 2026-01-06

**Summary:**

This paper presents Embodiment Grouping (EG), a morphology-aware approach for cross-embodiment offline RL. The authors observe that gradient conflicts arise when training shared policies across diverse morphologies. They measure embodiment similarity via the Fused Gromov–Wasserstein (FGW) distance, which jointly considers kinematic structure and behavior embeddings, and use this to cluster robots into morphology-aligned groups for group-wise policy training.


The paper got scores 6,6,6,4.  The main concern of the Reviewer umLy  (score 4) is the limited set of rewards/tasks and the narrow focus of the IQL algorithm and its variants.

The authors provide a detailed response to all reviewers and may address the reviewers' concerns.

**Reviewer Concerns:**

The main concern of the Reviewer umLy  (score 4) is the limited set of rewards/tasks and the narrow focus of the IQL algorithm and its variants.

The authors provide a detailed response to all reviewers and may address the reviewers' concerns.

**Reviewer Scores:**

The paper got scores 6,6,6,4.  After the rebuttal, the scores may reach above the acceptance threshold.

---

### Decision · Program_Chairs · 2026-01-26

Accept (Poster)